# Polo-like kinase-dependent phosphorylation of the synaptonemal complex protein SYP-4 regulates double-strand break formation through a negative feedback loop.

Saravanapriah Nadarajan[1], Talley J Lambert[2], Elisabeth Altendorfer[1], Jinmin Gao[1], Michael D Blower[1,3], Jennifer C Waters[2], Monica P Colaiácovo[1]*

[1]Department of Genetics, Harvard Medical School, Boston, United States; [2]Department of Cell Biology, Harvard Medical School, Boston, United States; [3]Department of Molecular Biology, Massachusetts General Hospital, Boston, United States

**Abstract** The synaptonemal complex (SC) is an ultrastructurally conserved proteinaceous structure that holds homologous chromosomes together and is required for the stabilization of pairing interactions and the completion of crossover (CO) formation between homologs during meiosis I. Here, we identify a novel role for a central region component of the SC, SYP-4, in negatively regulating formation of recombination-initiating double-strand breaks (DSBs) via a feedback loop triggered by crossover designation in *C. elegans*. We found that SYP-4 is phosphorylated dependent on Polo-like kinases PLK-1/2. SYP-4 phosphorylation depends on DSB formation and crossover designation, is required for stabilizing the SC in pachytene by switching the central region of the SC from a more dynamic to a less dynamic state, and negatively regulates DSB formation. We propose a model in which Polo-like kinases recognize crossover designation and phosphorylate SYP-4 thereby stabilizing the SC and making chromosomes less permissive for further DSB formation.

*For correspondence: mcolaiacovo@genetics.med.harvard.edu

**Competing interests:** The authors declare that no competing interests exist.

## Introduction

Meiosis is a specialized cell division program during which the diploid germ cell genome is halved to generate haploid gametes, and therefore it is critical for sexual reproduction. This halving is achieved by following one round of DNA replication with two consecutive rounds of cell division where homologs segregate away from each other at meiosis I, and sister chromatids segregate at meiosis II. Accurate chromosome segregation at meiosis I depends on several earlier key steps including pairing and the assembly of the synaptonemal complex (SC) between homologous chromosomes (homologs), formation of programmed DNA double-strand breaks (DSBs) and the repair of a subset of these DSBs as interhomolog crossover (CO) recombination events (*Page and Hawley, 2003*).

The SC plays a conserved and central role during meiosis. The assembly of the SC is required for the stabilization of pairing interactions between the homologs and for interhomolog CO formation (*Nag et al., 1995*; *Storlazzi et al., 1996*; *Page and Hawley, 2001*; *MacQueen et al., 2005*; *Colaiácovo et al., 2003*; *de Vries et al., 2005*; *Smolikov et al., 2009*, *Smolikov et al., 2007b*). While the components of the SC do not share a high degree of sequence conservation between

**eLife digest** The majority of DNA in animal cells is stored in structures called chromosomes. Most cells contain two sets of chromosomes, one inherited from the mother and one from the father. Sperm and egg cells, however, contain only a single set of chromosomes. A specialized type of cell division called meiosis generates these cells. During meiosis, the chromosomes in a cell replicate to produce a cell that contains four copies of each chromosome. The equivalent chromosomes from the mother and the father are initially kept close together by a zipper-like structure called the synaptonemal complex. This allows the chromosomes to exchange segments of DNA, before the cell divides twice in successive rounds to produce four cells, each containing one set of chromosomes.

Severing both of the DNA strands that make up a DNA molecule forms what is known as a double-strand break. To exchange DNA segments with another chromosome, double-strand breaks that form in the DNA of one chromosome are repaired in a process known as crossover formation. Only a subset of the double-strand breaks are designated to be repaired by crossover formation, but at least one crossover needs to form between each chromosome pair. This generates diversity and ensures that the chromosomes separate correctly at the first cell division.

Since the synaptonemal complex holds equivalent chromosomes close together it ensures that at least some of the breaks are repaired by crossover formation. Nadarajan et al. have now investigated how a chemical modification called phosphorylation affects how the synaptonemal complex behaves in the roundworm *Caenorhabditis elegans*. A combination of genetic and cell-based approaches revealed that enzymes called polo-like kinases phosphorylate one of the proteins – called SYP-4 – that makes up the synaptonemal complex. This phosphorylation occurs after double-strand break sites have been designated to become crossovers.

The synaptonemal complex is normally a dynamic structure, with the proteins that it consists of being continuously replaced. However, Nadarajan et al. found that phosphorylating SYP-4 made the synaptonemal complex less dynamic than it had previously been, which prevented further double-strand breaks from forming.

Polo-like kinases are found in many organisms, from yeast to humans. Further work is now needed to investigate whether polo-like kinases phosphorylate the synaptonemal complex – and hence prevent the continuous formation of double-strand breaks – in animals such as mice and humans. This is important because failing to shut down the formation of double-strand breaks can result in cancer, infertility, miscarriages and birth defects in humans.

---

species, this tripartite proteinaceous structure is conserved at the ultrastructural level (*Colaiácovo, 2006*). The SC is comprised of lateral element components that assemble along chromosome axes and central region components that bridge the gap between each pair of axes, much like the steps on a ladder. In the nematode *C. elegans*, there are four central region components of the SC, SYP-1, SYP-2, SYP-3 and SYP-4, which begin to localize between the homologs in the leptotene/zygotene stage of meiotic prophase and are fully assembled throughout the interface between homologs by pachytene (*MacQueen et al., 2002*; *Colaiácovo et al., 2003*; *Smolikov et al., 2009*, *Smolikov et al., 2007b*, *Smolikov et al., 2007a*). However, the SC is not static and has been shown to be a dynamic structure that undergoes continuous turnover. In budding yeast, Zip1, the central region component of the SC, has been shown to be continuously incorporated into a fully assembled SC in a non-uniform spatial pattern influenced by recombination (*Voelkel-Meiman et al., 2012*). In *C. elegans*, recent studies have shown that the central region of the SC contains mobile subunits (*Rog et al., 2017*) and that the SC persists in a more dynamic state in the absence of DSBs, leading to the suggestion that CO-committed intermediates may stabilize the SC (*Machovina et al., 2016*). However, how these changes in SC dynamics are regulated is not well understood.

Interhomolog CO formation is extremely important not only for producing genetic diversity but also for generating physical linkages (chiasmata) between homologs, which are essential for the proper bi-orientation and separation of homologs at meiosis I (*Page and Hawley, 2003*). Meiotic recombination initiates with the formation of programmed DSBs induced by the conserved

topoisomerase-like protein Spo11/SPO-11 (*Keeney et al., 1997*; *Bergerat et al., 1997*; *Dernburg et al., 1998*). DSBs are made in excess to ensure that at least one CO (obligate CO) is established for each pair of homologs (*Yokoo et al., 2012*; *Jones, 1984*). However, the engagement of homologous chromosomes during recombination and/or SC assembly has been proposed to turn off further programmed DSB formation (reviewed in *Keeney et al., 2014*). One possibility is that once a DSB is designated to become a CO a feedback mechanism turns off further programmed DSBs from forming. An alternative, albeit non-mutually exclusive possibility, is that SC formation may result in structural changes along the chromosomes which suppress further DSB formation. The molecular basis for transmission of this feedback regulation remains an open question.

Here, we provide a mechanistic basis for how the switch in SC dynamics and the transmission of feedback regulation on further DSB formation work during meiosis in *C. elegans*. We show that the Polo-like kinases, PLK-1 and PLK-2, are critical components of this negative feedback loop that couple CO designation with DSB formation. We show that a central region component of the SC, SYP-4, is phosphorylated at Serine 269 in a PLK-1/2-dependent manner. Phosphorylation of SYP-4 coincides with the appearance of CO-promoting factor CNTD1/COSA-1 and is dependent on the formation of DSBs and CO designation/precursor formation, which in turn inhibits additional DSBs from being formed. Further, phosphorylation of SYP-4 by PLK-1/2 switches the central region of the SC from a more dynamic to a less dynamic state. Thus, we propose a model in which PLK-1/2 mediate the phosphorylation of SYP-4 in response to CO designation/precursor formation, which in turn stabilizes SC dynamics and prevents further DSB formation.

## Results

### PLK-2 localization on chromosomes during pachytene is dependent on the SYP proteins

*plk-2* encodes one of the three Polo-like kinases in *C. elegans* and is the ortholog of mammalian PLK1. PLK-2 localizes to chromosome-associated patches at the nuclear periphery of leptotene/zygotene nuclei, which correspond to the regions of the chromosomes referred to as pairing centers that are tethered to the nuclear envelope at this stage, and relocalizes to synapsed chromosomes in pachytene nuclei (*Labella et al., 2011*; *Harper et al., 2011*). To test whether PLK-2 localization on chromosomes is dependent on central region components of the SC, we analyzed the localization of PLK-2 in the s*yp-1(me17)* and *syp-4(tm2713)* null mutants, which fail to synapse. We found that PLK-2 localized to the chromosome-associated patches at the nuclear periphery in leptotene/zygotene stage nuclei, but we did not detect PLK-2 signal on chromosomes in pachytene nuclei in *syp-1* and *syp-4* mutants (*Figure 1*). In *C. elegans,* PLK-1 functions redundantly with PLK-2 and it can partially substitute for the function of PLK-2 during pairing and synapsis of homologous chromosomes (*Nishi et al., 2008*; *Harper et al., 2011*; *Labella et al., 2011*). To determine whether PLK-1 also localized to synapsed chromosomes, we examined the immunolocalization of PLK-1 in wild type germlines. While PLK-1 is observed on chromosome-associated aggregates at the nuclear periphery during leptotene/zygotene as expected, we did not detect any PLK-1 signal in pachytene stage nuclei even in the absence of PLK-2, although we cannot rule out the possibility that PLK-1 signal could be below threshold levels of detection by immunofluorescence (*Figure 1—figure supplement 1*). Given that all of the four SYP proteins are interdependent on each other for their localization and for SC formation (*Colaiácovo et al., 2003*; *Smolikov et al., 2009*, *Smolikov et al., 2007a*), we infer from our analysis of the *syp-1* and *syp-4* mutants that PLK-2 localization to chromosomes during pachytene is dependent on all four SYP proteins.

### SYP-4 phosphorylation is dependent on the Polo-like kinases PLK-1 and PLK-2

To determine whether the SYP proteins are potential targets for phosphorylation by the Polo-like kinases, we examined the SYP-1/2/3/4 proteins for the presence of potential PLK phosphorylation sites using the phosphorylation site prediction programs GSP-polo and PHOSIDA (*Liu et al., 2013*; *Gnad et al., 2011*, *2007*). We identified S269 in SYP-4 as a potential Polo-like kinase phosphorylation site (*Figure 2A*). To test whether the S269 site is phosphorylated in vivo, we generated a phosphopeptide antibody against the SYP-4 S269 site. Phospho-specificity of the pSYP-4 antibody was

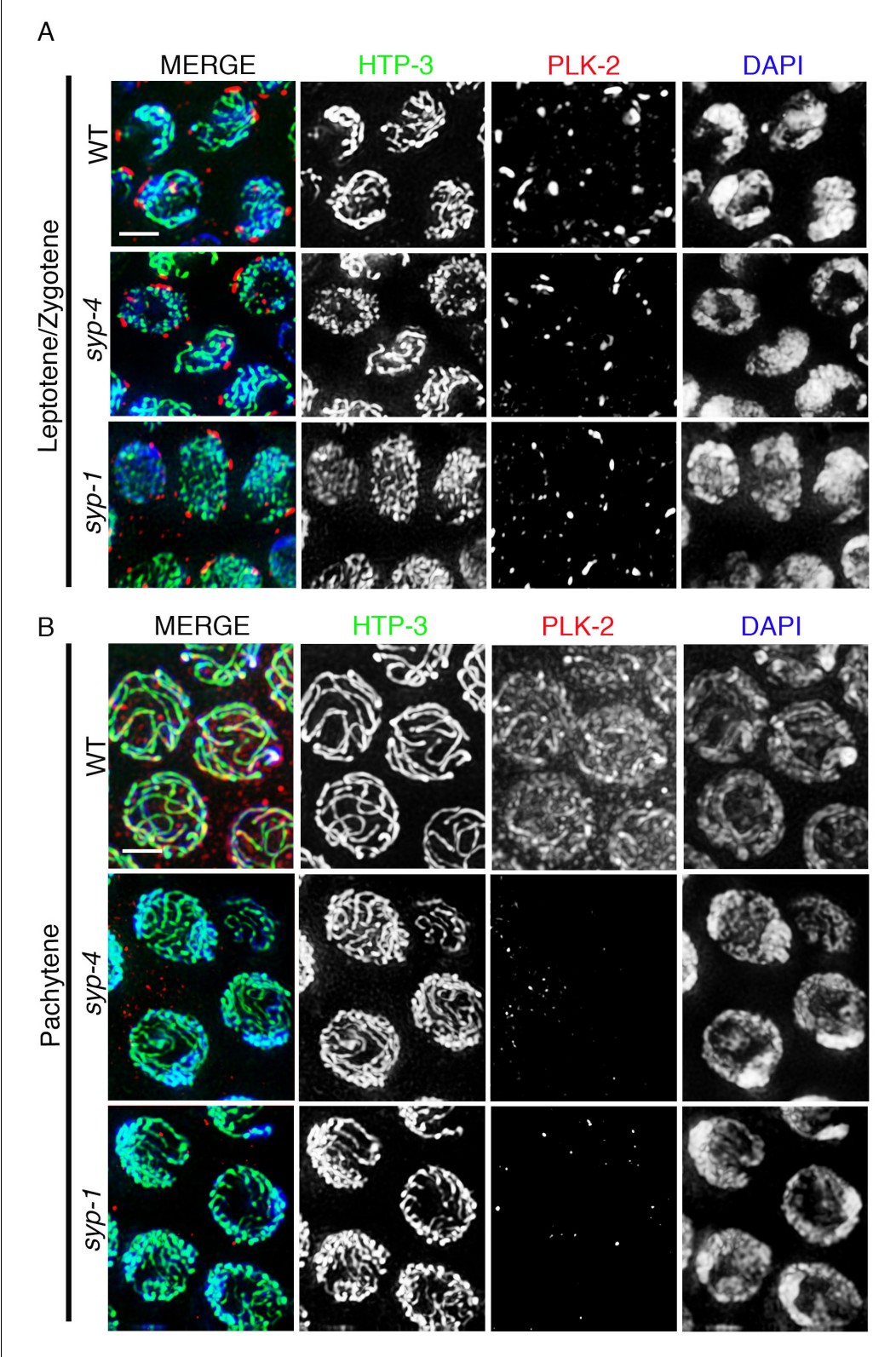

**Figure 1.** PLK-2 localization on synapsed chromosomes is dependent on the SYP proteins. High-magnification images of leptotene/zygotene (**A**) and pachytene stage (**B**) nuclei from wild type, *syp-4,* and *syp-1* mutant gonads stained with HTP-3 (green), PLK-2 (red), and DAPI (blue). (**A**) PLK-2 is observed localizing to aggregates at the nuclear periphery during leptotene/zygotene in synapsis-defective mutants. These aggregates, which have been previously shown to correspond to the pairing centers (*Labella et al., 2011*; *Harper et al., 2011*), are larger in wild type than in the *syp-1* and

*Figure 1 continued on next page*

*Figure 1 continued*

*syp-4* mutants given that the pairing centers of homologous chromosomes are not held in close juxtaposition in these mutants as frequently as in wild type at this stage in meiosis. (B) The more extensive localization of PLK-2 observed along chromosomes during pachytene in wild type is lost in *syp-1* and *syp-4* mutants. The thinner continuous tracks of HTP-3 staining observed in *syp* null mutants are due to HTP-3 localizing to unsynapsed axes. 27, 17, and 15 gonad arms were analyzed for wild type, *syp-4*, and *syp-1*, respectively. Scale bar, 3 μm.
The following figure supplement is available for figure 1:

**Figure supplement 1.** PLK-1 localization in the *C. elegans* germline.

confirmed by lack of pSYP-4 signal in the *syp-4(S269A)* phosphodead mutant (*Figure 2D*). Affinity purified phospho-specific SYP-4 antibody signal (pSYP-4) was detected on chromosomes starting at early pachytene, even though SYP-4 localizes at the interface between homologous chromosomes earlier, starting at the leptotene/zygotene stage (*Smolikov et al., 2009*). pSYP-4 was first observed as foci and short stretches on chromosomes in early pachytene nuclei and then fully colocalized with SYP-1 throughout the full length of the chromosomes by mid-pachytene (*Figure 2B and C*). In late pachytene, as the SC starts to disassemble, pSYP-4 signal was also lost from most of the interface between homologs and was restricted to a single portion of each bivalent referred to as the short arm of the bivalent, which also retains residual SC proteins until later in prophase I and which we have previously shown to correspond to the shortest distance from the position of the single CO event to a chromosome end (*Figure 2B and C*; (*Nabeshima et al., 2005*). To test whether the S269 site on SYP-4 is phosphorylated in a PLK-2-dependent manner in vivo, we immunostained *plk-2 (ok1936)* gonad arms with the pSYP-4 antibody. Although we observed a strong reduction in the pSYP-4 signal compared to wild type, there was residual phosphorylation of SYP-4 in the *plk-2* mutant. Since PLK-1 can substitute for the function of PLK-2 during the pairing of homologous chromosomes, we examined whether SYP-4 was phosphorylated in the absence of both PLK-1 and PLK-2. We found that pSYP-4 signal, but not SYP-4, was completely lost upon depletion of *plk-1* by RNAi in the *plk-2* mutant (*Figure 2D* and *Figure 2—figure supplement 1*). Taken together, our data suggest that SYP-4 is phosphorylated at the S269 site and that its phosphorylation is dependent on Polo-like kinases PLK-1 and PLK-2.

## Phosphorylation of SYP-4 is dependent on crossover precursor formation

Although the SYP proteins load on chromosomes starting in leptotene/zygotene, SYP-4 is phosphorylated starting at early pachytene, which coincides with the time when CO precursor markers are observed (*Yokoo et al., 2012*). We next tested the hypothesis that SYP-4 is phosphorylated in response to CO precursor formation. As we predicted, phosphorylation of SYP-4 coincided with the appearance of pro-crossover factor GFP::COSA-1 foci (*Figure 3A*). In early pachytene, GFP::COSA-1 foci were first observed only in some, but not on all, of the six pairs of homologous chromosomes. Strikingly, at that same stage, phosphorylated SYP-4 was first observed on 76% of chromosomes (n = 78) with GFP::COSA-1 foci, whereas a low percentage of chromosomes showed either only pSYP-4 signal (5.8%; n = 6) or COSA-1 foci (18.4%; n = 19) (*Figure 3A* and *Figure 3—figure supplement 1*). This result suggests that SYP-4 may be phosphorylated in response to CO precursor formation.

To further examine the link between SYP-4 phosphorylation and recombination, we examined phosphorylation of SYP-4 in the *spo-11(ok79)* null mutant. We found that pSYP-4 was largely abrogated in *spo-11* mutants except on a single chromosome track in 19.7% (n = 126) of mid to late pachytene nuclei (*Figure 3B* and *Figure 3—figure supplement 2*). Moreover, a COSA-1::GFP focus was observed on each of these chromosomes exhibiting pSYP-4 signal (*Figure 3—figure supplement 3A*). These residual COSA-1 foci and SYP-4 phosphorylation tracks may reflect recombination initiated from spontaneous DSBs or other DNA lesions, and is similar to the observation of SPO-11-independent COSA-1 foci in late pachytene nuclei made by *Pattabiraman et al., 2017*. To determine which chromosome exhibited the pSYP-4 track in *spo-11* mutants, we identified the X chromosome by HIM-8 staining and chromosomes III and V by fluorescence in situ hybridization. We found that 46% of pSYP-4 tracks corresponded to the X chromosome, 11.6% to chromosome V and 9.5%

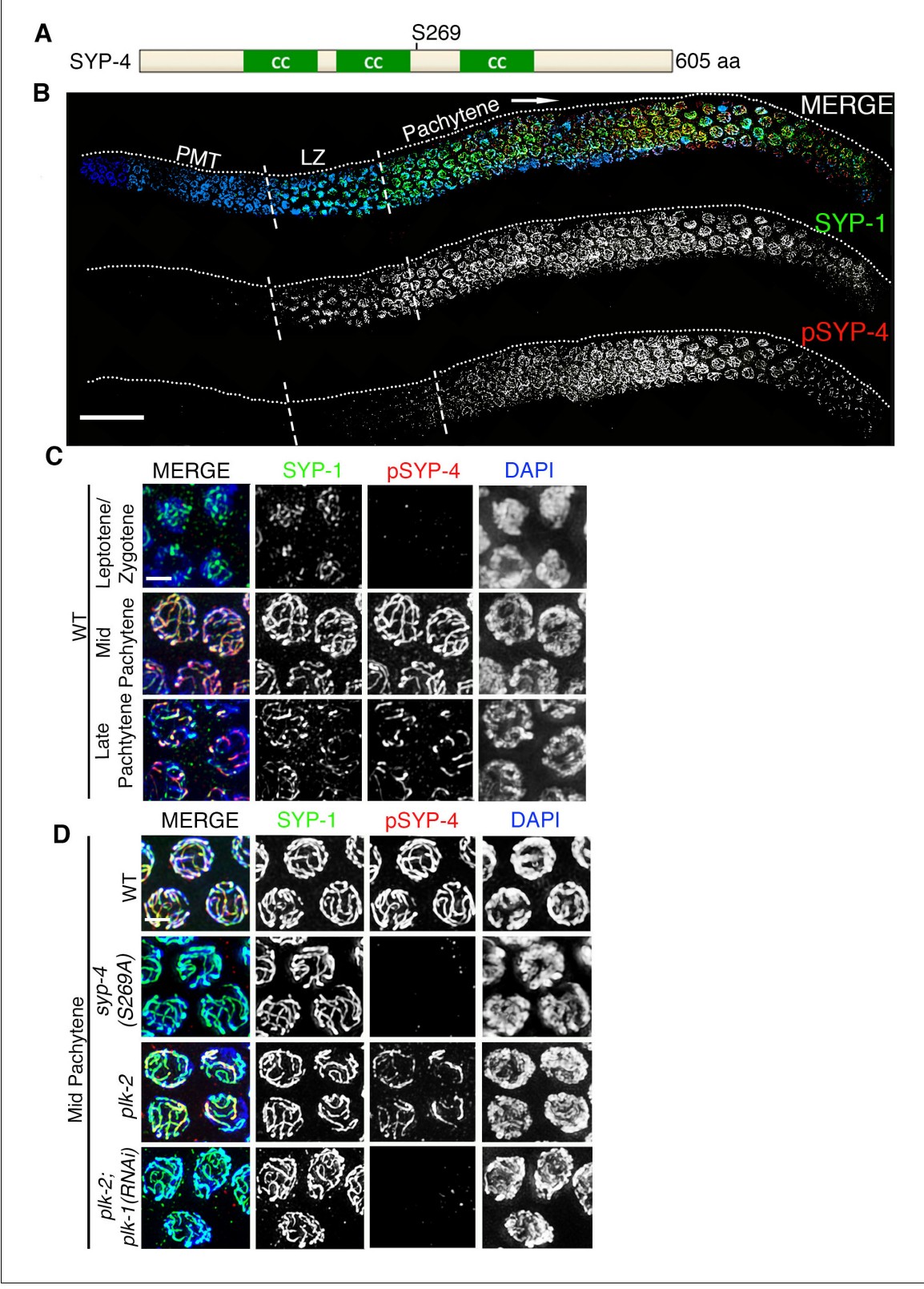

**Figure 2.** Phosphorylation of SYP-4 at the S269 site is dependent on Polo-like kinase. (**A**) Schematic representation of the SYP-4 protein with the predicted Polo-like kinase phosphorylation site (S269) indicated. CC indicates the coiled-coil domains and aa indicates amino acid. (**B**) Low-magnification images of whole-mounted gonads depicting SYP-1 (green) and phosphorylated SYP-4 (pSYP-4; red) localization in wild type. pSYP-4 signal is observed starting at early-pachytene although SC assembly, as observed here by SYP-1 immunostaining, starts earlier at the leptotene/

*Figure 2 continued on next page*

*Figure 2 continued*

zygotene stage. PMT (premeiotic tip) and LZ (leptotene/zygotene). (**C**) High-magnification images of wild type germline nuclei at the indicated stages stained with DAPI (blue), anti-SYP-1 (green) and anti-pSYP-4 (red). 16 gonad arms were analyzed for wild type (**B–C**). Phosphorylated SYP-4 signal is absent at leptotene/zygotene, observed colocalizing with SYP-1 in mid-pachytene and acquiring a similar restricted localization as SYP-1 during the disassembly of the SC starting at late pachytene. (**D**) High-magnification images of mid-pachytene nuclei from wild type, *plk-2(ok1936), plk-2(ok1936); plk-1(RNAi),* and *syp-4(S269A)* mutants stained with DAPI (blue), anti-SYP-1 (green) and anti-pSYP-4 (red). Phosphorylated SYP-4 signal is absent in the *syp-4(S269A)* mutant indicating specificity of the phospho-specific antibody. Phosphorylated SYP-4 signal is reduced in mid-pachytene nuclei in *plk-2* mutants and absent at that stage in *plk-2; plk-1(RNAi)* germlines. Note that an uneven SYP-1 signal intensity is observed along chromosomes in *plk-2; plk-1(RNAi)* germlines, but pSYP-4 signal is not detected even on chromosomes with strong SYP-1 signal. 22, 17, 16, and 18 gonad arms were analyzed for wild type, *plk-2(ok1936), plk-2(ok1936); plk-1(RNAi),* and *syp-4(S269A)* mutants, respectively. Scale bar, 20 μm for (**B**) and 3 μm for (**C**) and (**D**).

The following figure supplement is available for figure 2:

**Figure supplement 1.** RT-PCR analysis of *plk-1* knockdown by RNAi.

to chromosome III (*Figure 3—figure supplement 3B and C*). The higher frequency of pSYP-4 tracks along the X chromosome could be due to its distinct chromatin structure and/or a result of it undergoing delayed replication compared to autosomes (*Bender et al., 2004*; *Kelly et al., 2002*; *Jaramillo-Lambert et al., 2007*), which might lead to a higher number of DNA lesions that could be processed for CO precursor formation during meiosis.

Introduction of exogenous DSBs by γ-irradiation restored SYP-4 phosphorylation, suggesting that the absence of SYP-4 phosphorylation in *spo-11* mutants was due to lack of DSBs and not due to the lack of the SPO-11 protein (*Figure 3B* and *Figure 3—figure supplement 1*). We observed a similar loss of phosphorylated SYP-4, except for a single chromosome track in <20% of nuclei, in mutants for genes involved in DSB formation (*spo-11, dsb-1,* and *dsb-2*), resection (*mre-11*) and strand invasion/exchange (*rad-51* and *rad-54*) during homologous recombination (*Figure 3* and *Figure 3—figure supplement 2*). However, the SYP-4 phosphorylation signal was completely absent in pachytene stage nuclei in the *zhp-3* and *cosa-1* pro-crossover defective mutants (*Figure 3* and *Figure 3—figure supplement 2*). Taken together, our data suggest that SYP-4 phosphorylation is dependent on CO precursor formation.

## Homologous chromosome pairing and meiotic progression are not altered in *syp-4* phosphodead and phosphomimetic mutants

Previous work has shown that Polo-like kinases regulate pairing in *C. elegans* (*Labella et al., 2011*; *Harper et al., 2011*). To test whether phosphorylation of SYP-4 regulates pairing, we analyzed pairing in *syp-4(S269A)* phosphodead and *syp-4(S269D)* phosphomimetic mutants. Analysis of homologous pairing for chromosome V by fluorescence in situ hybridization (FISH) and for the X chromosome using a HIM-8 antibody that recognizes the X-chromosome pairing center revealed no significant difference in pairing levels throughout the germline for these chromosomes in either the phosphodead or phosphomimetic mutants compared to wild type (*Figure 4A*). To test whether Polo-like kinases regulate meiotic progression through phosphorylation of SYP-4 at the S269 site, we analyzed the localization of phosphorylated SUN-1 with a SUN-1 S8 phospho-specific antibody in *syp-4(S269A)* phosphodead and *syp-4(S269D)* phosphomimetic mutants. SUN-1 encodes for a conserved inner nuclear envelope protein, which clusters at chromosome ends associated with the nuclear envelope, and its phosphorylation is dependent on CHK-2 and PLK-2 (*Woglar et al., 2013*). In wild type, SUN-1 S8 phosphorylation is observed upon entry into meiosis in leptotene/zygotene stage nuclei and its signal is no longer detected around mid-pachytene, except in a few nuclei (*Woglar et al., 2013*). We did not observe any significant difference in SUN-1 S8P phospho-specific antibody localization in either *syp-4(S269A)* phosphodead or *syp-4(S269D)* phosphomimetic mutants compared to wild type (*Figure 4B*). Altogether, our data indicate that meiotic pairing and early meiotic progression are not dependent on the phosphorylation of SYP-4.

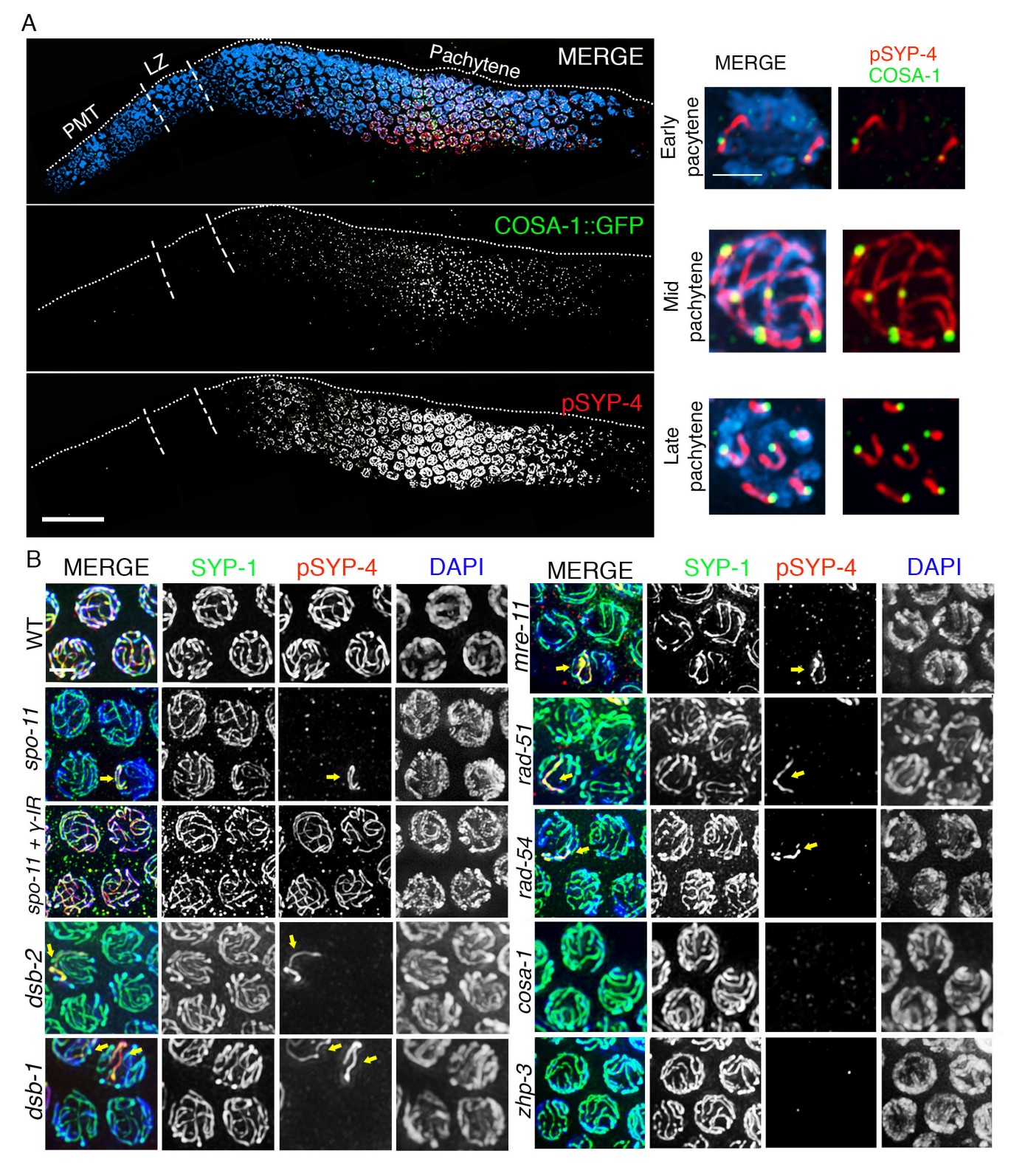

**Figure 3.** SYP-4 is phosphorylated in response to CO precursor formation. (**A**) Low-magnification image of a whole-mounted gonad from an animal expressing GFP::COSA-1 stained with anti-GFP (green), anti-pSYP-4 (red), and DAPI (blue). GFP::COSA-1 foci are detected from early to late pachytene, coinciding with the temporal window in which pSYP-4 signal is observed on chromosomes. Insets on the right are high-magnification images showing that stretches of pSYP-4 signal are first observed at early pachytene mainly on chromosomes that have a GFP::COSA-1 focus (top row of insets). pSYP-4

*Figure 3 continued on next page*

*Figure 3 continued*

signal is then observed continuously along the length of the chromosomes at mid-pachytene. Finally, pSYP-4 signal starts to be lost from some chromosome subdomains and is retained from the off-centered site of the CO event marked by GFP::COSA-1 through the shortest distance to one end of the chromosomes (this will later become the short arm of the bivalent; bottom row of insets). PMT (premeiotic tip) and LZ (leptotene/zygotene). Scale bar, 20 μm. 21 gonads were analyzed. (B) High-magnification images of pachytene stage nuclei stained with anti-SYP-1 (green), anti-pSYP-4 (red), and DAPI (blue) for the indicated genotypes. pSYP-4 signal is mostly lost in *spo-11*, *dsb-1*, *dsb-2*, *mre-11*, *rad-51* and *rad-54* mutants. Arrow points to the chromosome that still shows pSYP-4 localization in these mutants. The extensive localization of pSYP-4 along synapsed chromosomes can be rescued by exogenous induction of DSBs via γ-IR in *spo-11* mutants. pSYP-4 signal is not detected in pro-crossover mutants *cosa-1* and *zhp-3*. Scale bar, 20 μm for (A) and 3 μm for (B). At least 15 animals were examined for each genotype.

The following figure supplements are available for figure 3:

**Figure supplement 1.** Quantitation of colocalization between COSA-1::GFP foci and pSYP-4 signal on chromosomes in early pachytene.

**Figure supplement 2.** SYP-4 is phosphorylated in response to CO precursor formation.

**Figure supplement 3.** Assessing the residual SYP-4 phosphorylation observed in *spo-11* mutants.

## Polo-like kinases regulate DNA double-strand break formation through phosphorylation of SYP-4

*C. elegans* chromosomes exhibit strict CO interference and undergo a single CO per homologous chromosome pair (*Yokoo et al., 2012*; *Libuda et al., 2013*). Partial depletion of central region components of the SC has been shown to attenuate CO interference and elevate the number of COs observed between homologs (*Libuda et al., 2013*). To test the hypothesis that phosphorylation of SYP-4 regulates CO interference, we crossed GFP::COSA-1 into the *syp-4(S269A)* phosphodead and *syp-4(S269D)* phosphomimetic mutants and analyzed CO levels by counting the number of COSA-1 foci observed per nucleus. If phosphorylation of SYP-4 functions as a signal to promote CO interference and prevent more than one CO per chromosome, we would expect to see an increase in the number of COs per nucleus in the phosphodead mutants as that signal is no longer present. However, we did not observe a significant increase in the number of COs, as marked by COSA-1, per nucleus in *syp-4(S269A)* phosphodead mutants (*Figure 5A*). Similarly, we did not observe a significant decrease in the number of COs in *syp-4(S269D)* phosphomimetic mutants (*Figure 5A*). We observed a significant reduction in brood size in both *syp-4(S269A)* phosphodead and *syp-4(S269D)* phosphomimetic mutants compared to wild type (*Supplementary file 1*). However, we only saw 6–8% embryonic lethality accompanied by 0.3–1% males among the progeny from these mutants, suggesting only a mild increase of autosomal and X chromosome nondisjunction during meiosis in these mutants, which could stem in part from the <4% of oocytes observed having a reduced number of COSA-1 foci (*Supplementary file 1*). These results suggest that PLK-dependent phosphorylation of SYP-4 does not regulate CO interference.

Unprocessed meiotic DSBs can trigger germ cell apoptosis in late pachytene due to activation of a DNA damage checkpoint in the *C. elegans* germline (*Schumacher et al., 2001*). Our analysis revealed a significant increase in the levels of germ cell apoptosis in both phosphodead and phosphomimetic mutants (*Figure 5B*). To test whether this is due to a role for PLK-dependent phosphorylation of SYP-4 in regulating the progression of meiotic recombination, we examined the levels of RAD-51 foci, a protein involved in DNA strand invasion/exchange during DSB repair, in nuclei throughout the germline of the *syp-4(S269A)* phosphodead and *syp-4(S269D)* phosphomimetic mutants. We observed a significant increase in the number of RAD-51 foci per nucleus starting at mid pachytene in *syp-4(S269A)* phosphodead mutants compared to wild type, and the levels of foci remained higher than wild type through late pachytene (*Figure 5D* and *Figure 5—figure supplement 1*). In contrast, the number of RAD-51 foci observed in early and mid-pachytene nuclei was significantly lower than in wild type in *syp-4(S269D)* phosphomimetic mutants (*Figure 5D*). These results suggest that PLK-dependent phosphorylation of SYP-4 regulates progression of meiotic recombination. To distinguish whether this phosphorylation regulates DSB formation or repair we quantitated the levels of RAD-51 foci observed per nucleus throughout the germline in the absence of RAD-54. Depletion of *rad-54* traps RAD-51 association at DSB sites, which allows for the scoring

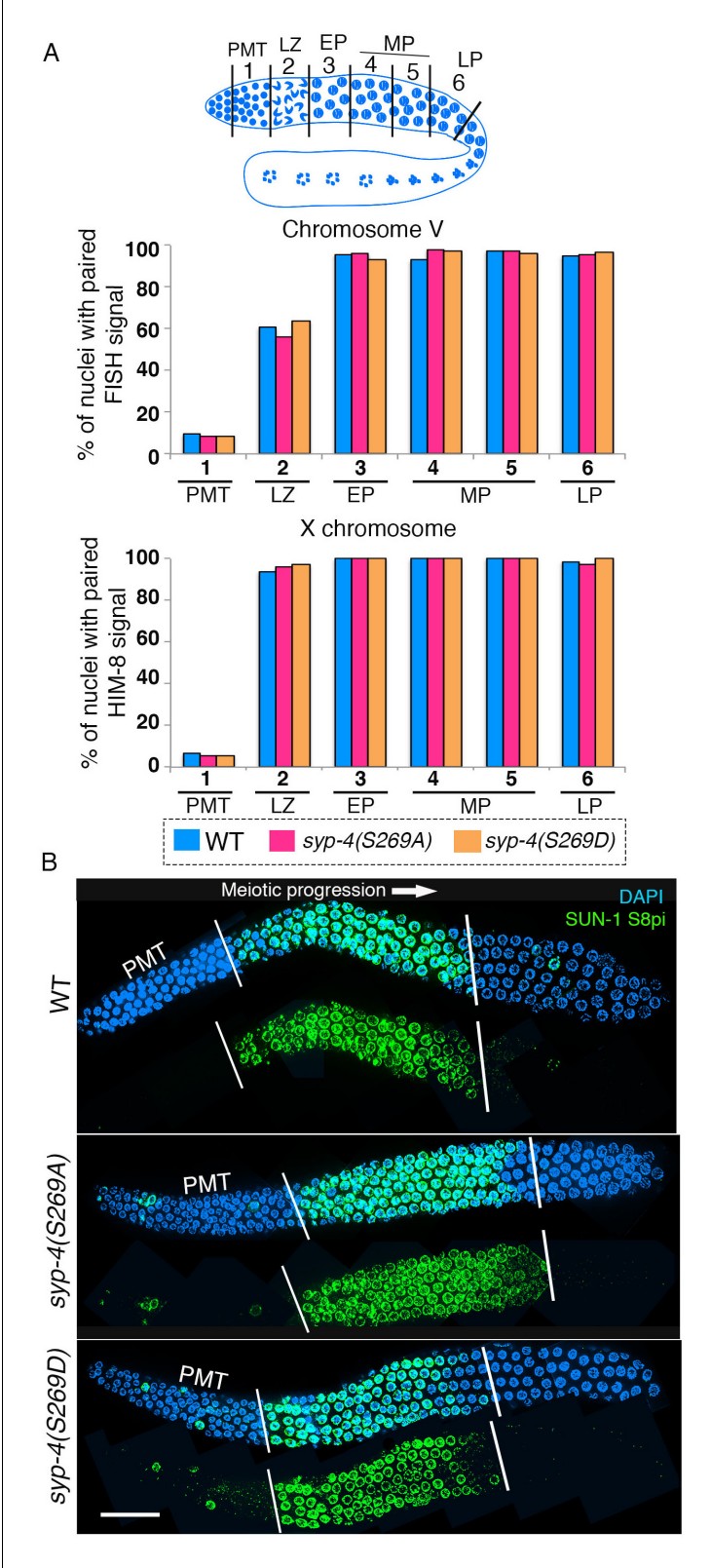

**Figure 4.** SYP-4 phosphorylation at the S269 site does not affect pairing or SUN-1 S8pi localization. Homologous pairing for chromosomes V and X was analyzed using FISH and immunostaining against the X chromosome pairing center protein HIM-8, respectively. (**A**) Schematic representation of the *C. elegans* germline indicating the different zones scored for homologous chromosome pairing. Graphs depict no statistically significant difference in
*Figure 4 continued on next page*

*Figure 4 continued*

the percentage of nuclei with paired FISH signals for chromosome V and paired HIM-8 signals for the X-chromosome in the germline of the indicated mutants compared to wild type (signals were scored as paired when separated by ≤0.75 µm). X-axes indicate the position along the germline. PMT- premeiotic tip, L/Z- leptotene/zygotene, EP- early pachytene, MP- mid pachytene, and LP- late pachytene. Six gonad arms were analyzed for each genotype. (B) Low-magnification images of whole-mounted gonads indicate no difference in the length of the germline region stained with anti-SUN-1 S8pi (green) for wild type (top), *syp-4(S269A)* phosphodead (middle), and *syp-4(S269D)* phosphomimetic mutants (bottom). Left white vertical bar indicates entrance into meiosis and right vertical bar indicates end of zone where SUN-1 S8pi signal is detected. Scale bar, 15 µm. >18 gonad arms were analyzed for each genotype.

The following source data is available for figure 4:

**Source data 1.** Numerical data for the percentage of nuclei with paired FISH signals for chromosome V and paired HIM-8 signals for the X-chromosome in the germline of the indicated mutants compared to wild type shown in *Figure 4A*.

---

of the total number of DSBs repaired via a RAD-51 intermediate (*Mets and Meyer, 2009*). We observed significantly elevated levels of RAD-51 foci in *syp-4(S269A)* phosphodead mutant germlines upon depletion of *rad-54* by RNAi compared to *rad-54(RNAi)* in an otherwise wild type background (*Figure 5E*). In contrast, *syp-4(S269D)* phosphomimetic mutant germlines displayed significantly reduced RAD-51 foci levels upon *rad-54(RNAi)* compared to control (*Figure 5E*). These results indicate that phosphorylation of SYP-4 at S269 regulates DSB formation. Altogether, these data support a role for PLK-dependent phosphorylation of SYP-4 in negatively regulating DSB formation in the *C. elegans* germline.

## Phosphorylation of SYP-4 regulates the dynamics of the central region of the SC

In *C. elegans*, the assembly of the SC is independent of DSB formation (*Dernburg et al., 1998*). To understand the role of PLK-dependent phosphorylation on the SC structure, we analyzed the immunolocalization of the lateral element protein HTP-3, and all four central region components of the SC, SYP-1/2/3/4, in *syp-4(S269A)* phosphodead and *syp-4(S269D)* phosphomimetic mutants. We did not observe obvious defects in either SC assembly or maintenance in either mutant background (*Figure 6—figure supplement 1*). Moreover, western blot analysis revealed no change in SYP-1 or SYP-3 protein levels in whole worm lysates from either *syp-4(S269A)* phosphodead or *syp-4(S269D)* phosphomimetic mutants compared to wild type (we were precluded from examining SYP-2 and SYP-4 in this way since the available antibodies do not work on westerns) (*Figure 6—figure supplement 1*).

To test whether phosphorylation of SYP-4 affects the dynamics of the SC central region we performed Fluorescence Recovery After Photobleaching (FRAP) in live animals expressing SYP-3::GFP in an otherwise wild-type background as well as in the *syp-4(S269A)* phosphodead and *syp-4(S269D)* phosphomimetic mutants. We measured the extent of recovery of the fluorescence signal post-photobleaching for nuclei in the leptotene/zygotene stage, where SYP-4 is not phosphorylated, and at mid-pachytene, where SYP-4 is phosphorylated on all chromosomes in wild type. First, we observed that the majority of the SYP-3::GFP signal was recovered within 20 min after photobleaching in wild type nuclei. However, the extent of recovery of the fluorescence signal was higher in leptotene/zygotene nuclei compared to mid-pachytene nuclei in wild type (p<0.0001 by the Dunn's multiple comparisons test), suggesting that the central region of the SC transitions from a more dynamic state into a less dynamic or more stable state as meiotic prophase progresses (*Figure 6*, *Supplementary file 2*, *Video 1* and *Video 2*. In contrast, in *syp-4(S269A)* phosphodead mutants, the extent of recovery for the SYP-3::GFP signal was high in both leptotene/zygotene and mid-pachytene stage nuclei, and similar at mid-pachytene to that measured for leptotene/zygotene nuclei in wild type (*Figure 6* and *Supplementary file 2*). This result suggests that phosphorylation of SYP-4 can alter the dynamics of the central region of the SC by stabilizing it. This is further supported by our observation of a lower extent of recovery of the SYP-3::GFP signal after photobleaching in leptotene/zygotene nuclei in *syp-4(S269D)* phosphomimetic mutants compared to that observed for wild-

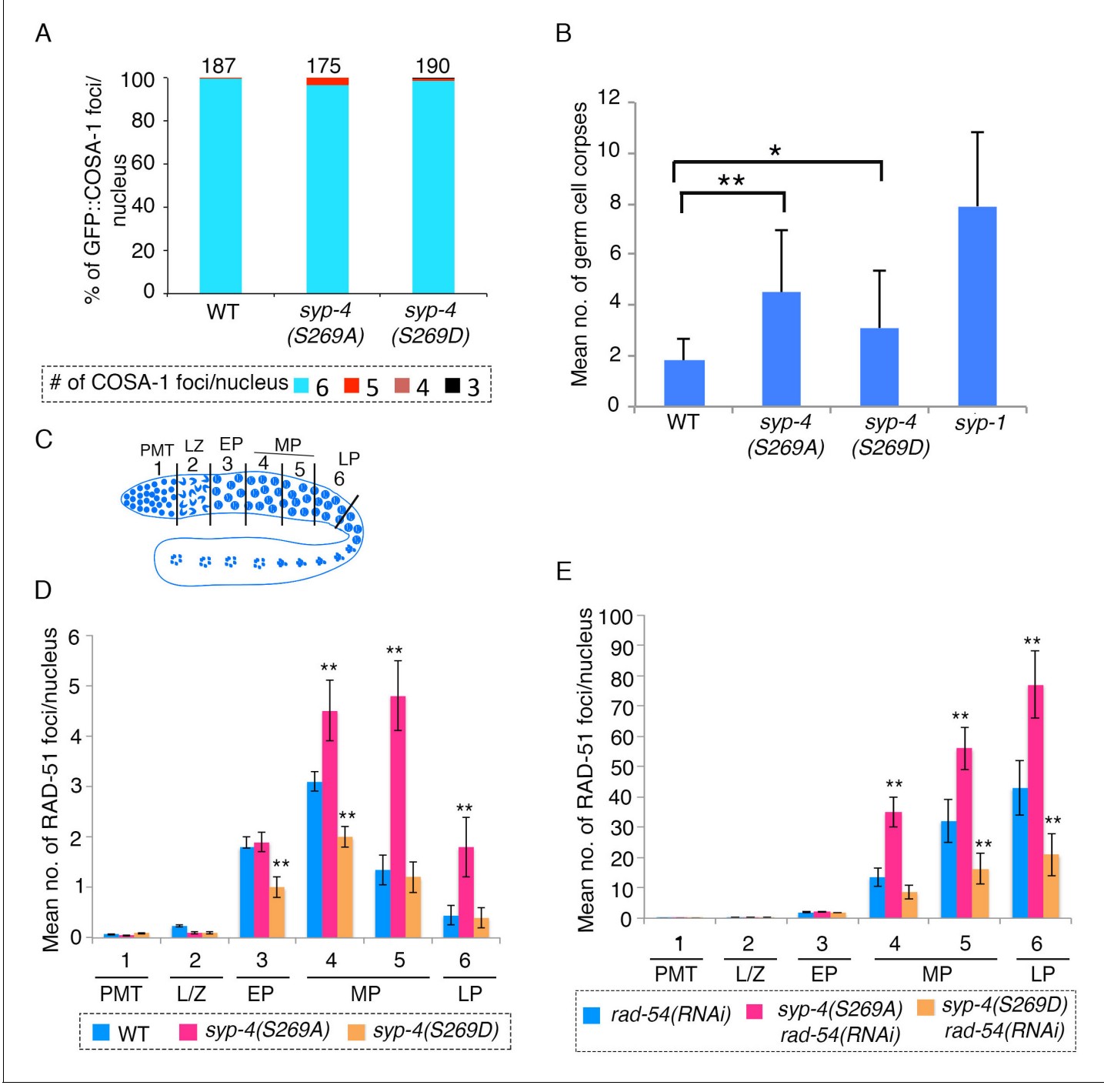

**Figure 5.** SYP-4 phosphorylation at the S269 site by Polo-like kinase regulates DSB formation. (**A**) Graph depicts no statistically significant difference in the percentage of GFP::COSA-1 foci scored per nucleus for either the *syp-4(S269A)* or *syp-4(S269D)* mutants compared to wild type. Color code at the bottom indicates the number of GFP::COSA-1 foci observed per nucleus. Number on the top of the histogram bars represent the total number of nuclei scored for each genotype. (**B**) Graph depicts the increase in the mean number of germ cell corpses detected for both *syp-4(S269A)* and *syp-4 (S269D)* mutants compared to wild type (\*\*p<0.0001 and \*p<0.0008, respectively, by the two-tailed Mann-Whitney test, 95% C.I.). The *syp-1* mutant was used as a positive control given its elevated levels of germ cell apoptosis. 56, 72, 61, and 40 animals were analyzed for wild type, *syp-4(S269A)*, *syp-4 (S269D)* and *syp-1(me17)* mutants, respectively. (**C**) Schematic representation of the *C. elegans* germline indicating the different zones scored for the number of RAD-51 foci/nucleus. (**D**) Histogram depicts the increase in the mean number of RAD-51 foci observed per nucleus in the germlines of *syp-4 (S269A)* mutants compared to wild type and the decrease in the mean number of RAD-51 foci observed *in syp-4(S269D)* mutants compared to wild type. (**E**) Similar analysis as in (**D**) except it is performed in a *rad-54(RNAi)* background allowing for a quantification of the total number of DSBs

*Figure 5 continued on next page*

*Figure 5 continued*

observed in each indicated genotype. This analysis reveals that the altered numbers of RAD-51 foci are due to increased levels of DSBs in *syp-4(S269A)* mutants and decreased levels of DSBs in *syp-4(S269D)* mutants. (**D–E**) X-axes indicate the position along the germline. PMT- premeiotic tip, L/Z- leptotene/zygotene, EP- early pachytene, MP- mid-pachytene, and LP- late pachytene. Six gonad arms were analyzed for each genotype. ** Indicates p<0.0001 by the two-tailed Mann-Whitney test, 95% C.I.

The following source data and figure supplement are available for figure 5:

**Source data 1.** Numerical data for the percentage of GFP::COSA-1 foci scored per nucleus for either the *syp-4(S269A)* or *syp-4(S269D)* mutants compared to wild type shown in ***Figure 5A***.

**Source data 2.** Numerical data used to calculate the mean number of germ cell corpses for both *syp-4(S269A)* and *syp-4(S269D)* mutants compared to wild type shown in ***Figure 5B***.

**Source data 3.** Numerical data used to calculate the mean number of RAD-51 foci observed per nucleus throughout zones 1–6 from whole mounted gonads of the indicated genotypes shown in ***Figure 5D and E***.

**Figure supplement 1.** RT-PCR showing depletion of *rad-54* by RNAi in wild type, *syp-4(S269A) rad-54(RNAi)* and *syp-4(S269D) rad-54(RNAi)* animals.

type nuclei at that stage (p<0.0001) (***Figure 6***). Specifically, the lower rate of recovery observed after photobleaching for the phosphomimetic mutant was similar for leptotene/zygotene and mid-pachytene nuclei and comparable to that observed in mid-pachytene stage nuclei for wild type (***Figure 6A*** and ***Supplementary file 2***). Altogether, these data suggest that PLK-dependent phosphorylation of SYP-4 in pachytene changes the central region of the SC from a more dynamic to a less dynamic, and therefore more stable, state, which in turn impinges on DSB formation.

## Discussion

We have discovered that Polo-like kinase-dependent phosphorylation of a central region component of the SC, SYP-4, negatively regulates DSB formation through a feedback loop in *C. elegans*. Our data suggest that Polo-like kinases may phosphorylate SYP-4 starting at early pachytene, which is consistent with when PLK-2 is observed localizing on the homologous chromosomes. Although SC assembly is DSB-dependent in many organisms, it is DSB-independent in *C. elegans* (***Loidl et al., 1994***; ***Baudat et al., 2000***; ***Romanienko and Camerini-Otero, 2000***; ***Dernburg et al., 1998***). Surprisingly, we found that DSB formation and recombination are required for phosphorylation of SYP-4. In mutants defective in DSB formation and recombination (*spo-11*, *dsb-1*, *dsb-2*, *mre-11*, *rad-51*, *rad-54*, *zhp-3*, and *cosa-1*), SYP-4 is no longer phosphorylated (except along one chromosome track for early recombination mutants) at the S269 site in pachytene nuclei. This raises an important question: does the Polo-like kinase-dependent phosphorylation of SYP-4 at the S269 site occur in response to recombination? We found that the appearance of foci for the pro-crossover factor COSA-1 coincides with the phosphorylation of SYP-4. Recent studies showed that central region components of the SC impose CO interference and prevent additional CO formation in response to CO designation (***Libuda et al., 2013***). We did not see any significant difference in CO number in *syp-4* phosphodead and phosphomimetic mutants compared to wild type, which rules out the possibility that phosphorylation of SYP-4 at S269 mediates the regulation of CO interference. Instead, we found that SYP-4 phosphorylation at S269 mediates the regulation of DSB formation. Recent studies, and this study, show that the central region of the SC exists in different dynamic states at different stages of meiotic prophase (***Voelkel-Meiman et al., 2012***; ***Machovina et al., 2016***): more dynamic at the leptotene/zygotene stage and less dynamic at the mid-pachytene stage in wild type. Our data suggest that the more dynamic state of the central region of the SC in leptotene/zygotene corresponds to the region in the germline where homologs are permissive for DSB formation, whereas the less dynamic state of the central region of the SC coincides with the stage of meiosis where homologs are less permissive to DSB formation. Thus we propose that upon CO designation/precursor formation, PLK-1/2 mediates the phosphorylation of the S269 site in SYP-4 (***Figure 7***). This phosphorylation event leads to the stabilization of the central region of the SC and makes it less dynamic. Therefore, phosphorylation of SYP-4 functions as a signaling mechanism to alter the SC

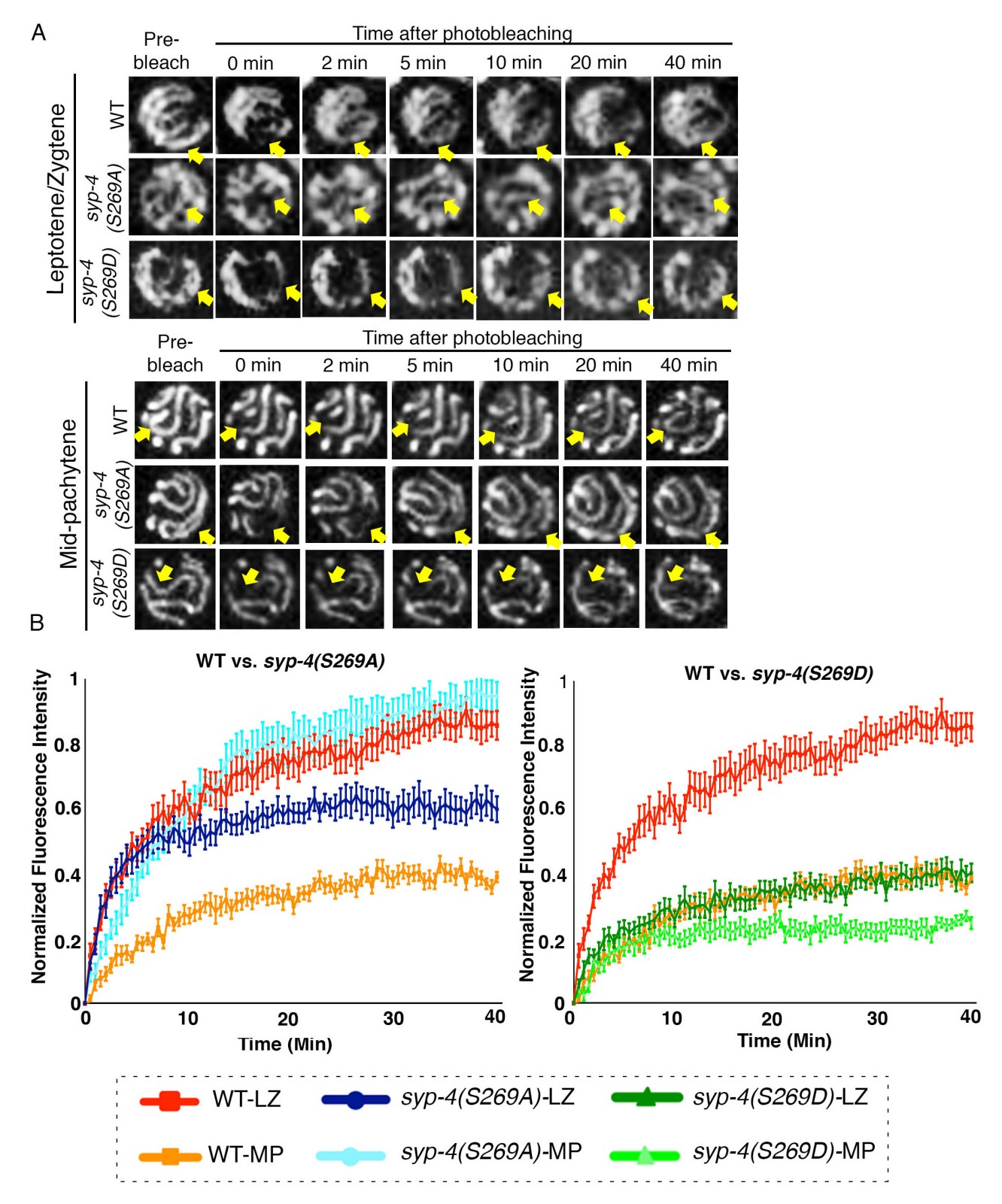

**Figure 6.** PLK-dependent phosphorylation of SYP-4 changes the SC central region from a more dynamic to a less dynamic state. (**A**) Representative images showing the GFP::SYP-3 fluorescence detected in leptotene/zygotene and mid-pachytene nuclei for the indicated genotypes during FRAP experiments. Arrows indicate the small region that was bleached and measured for fluorescence recovery on each nucleus. GFP::SYP-3 signal recovery after photobleaching during leptotene/zygotene is faster in both wild type and *syp-4(S269A)* mutants compared to *syp-4(S269D)* mutants. At mid-

*Figure 6 continued on next page*

*Figure 6 continued*

pachytene, while the rate of fluorescence signal recovery is slower compared to earlier prophase for wild type, the *syp-4(269A)* mutants continue to exhibit a rapid recovery rate comparable to that observed earlier in leptotene/zygote, while the *syp-4(S269D)* mutants continue to exhibit a slower recovery rate. (B) Graph showing quantitation of GFP::SYP-3 fluorescence recovery in leptotene/zygotene and mid-pachytene stage nuclei in wild type compared to *syp-4(S269A),* and *syp-4(S269D)* mutants. The total of number of nuclei measured in each group were: Mid-pachytene_WT: n = 20; Mid-pachytene _syp-4(S269D):* n = 18; Mid-pachytene _ *syp-4(S269A)*: n = 20; Leptotene/zygotene _WT: n = 28; Leptotene/zygotene_ *syp-4(S269D)*: n = 20; Leptotene/zygotene _ *syp-4(S269A)*: n = 21. Error bars represent standard error of the mean. Bar code in the bottom indicates the stage of the nuclei and genotype. LZ – leptotene/zygotene; MP – mid-pachytene.

The following source data and figure supplement are available for figure 6:

**Source data 1.** Final data, after corrections, used to make graph for FRAP analysis shown in *Figure 6B*.
**Source data 2.** Raw numerical data used for graph of FRAP analysis shown in *Figure 6B*.
**Figure supplement 1.** Phosphorylation of SYP-4 at S269 does not affect SC assembly and maintenance.

structure thereby inhibiting further DSB formation in *C. elegans*. Consistent with our model, the central region of the SC remains dynamic in later stages of meiosis in *syp-4* phosphodead mutants (this study) and in meiotic recombination defective mutants (*Machovina et al., 2016*; *Pattabiraman et al., 2017*). Moreover, the observation that the SC persists in a more dynamic state during pachytene in *spo-11* mutant animals (*Machovina et al., 2016*; *Pattabiraman et al., 2017*), despite the lack of programmed meiotic DSB formation in this mutant, rules out the possibility that the elevated DSB levels observed in *syp-4* phosphodead mutants cause the prolonged dynamic state of the SC.

Finally, in early pachytene, PLK-1/2-dependent phosphorylation of SYP-4 is observed along chromosomes that engage in CO designation in wild type worms. In *spo-11* mutants, we observed a single chromosome with a COSA-1::GFP focus colocalizing with a pSYP-4 track in approximately 20% of nuclei. This suggests that SYP-4 phosphorylation acts at the level of individual chromosomes, given that the phosphorylated state of SYP-4 at one chromosome does not affect or is independent of the state of SYP-4 phosphorylation on the other chromosomes.

## Novel role for conserved Polo-like kinase in ensuring CO formation

Polo-like kinase is a highly conserved kinase present from yeast to humans and has been shown to play multiple roles during both mitotic and meiotic cell divisions (*Archambault and Glover, 2009*). The yeast Polo-like kinase Cdc5 has been implicated in the resolution of double Holliday junctions in homologous recombination, promotion of CO formation, breakdown of the SC, co-orientation of sister chromatids, and loss of cohesion from chromosome arms at anaphase I (*Clyne et al., 2003*; *Lee and Amon, 2003*; *Brar et al., 2006*; *Attner et al., 2013*; *Sourirajan and Lichten, 2008*). In mouse spermatocytes, PLK1 has been shown to phosphorylate the central element proteins SYCP1 and TEX12 to promote SC disassembly and is required for exit from meiotic prophase (*Jordan et al., 2012*). In *C. elegans*, PLK-2 functions redundantly with PLK-1 to promote pairing and synapsis of homologous chromosomes in the leptotene/zygotene stage of meiosis (*Harper et al., 2011*; *Labella et al., 2011*). Our study reveals a novel role for PLK-1/2 in recognizing CO designation and generating an

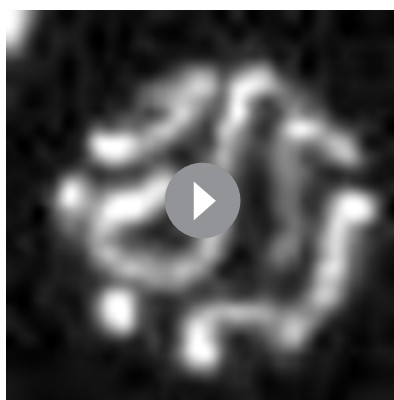

**Video 1.** Example of a nucleus included in the FRAP analysis for *Figure 6*. Mid-pachytene stage nuclei from wild type hermaphrodite animals expressing SYP-3:: GFP included for analysis for FRAP experiment in *Figure 6*. Playback time is 447x in real-time.

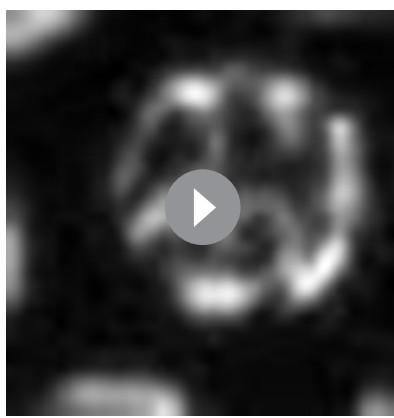

**Video 2.** Example of a nucleus excluded from FRAP analysis in *Figure 6*. Mid-pachytene stage nuclei from wild type hermaphrodite animals expressing SYP-3:: GFP excluded from the FRAP analysis since this nucleus is rotating on its own axis. Playback time is 447x in real-time.

inhibitory signal that prevents further DSB formation. The mechanism by which PLK-1/2 recognize CO designation remains to be determined.

## Feedback regulation of DSB formation and the dynamics of the central region of the SC

Even though DNA damage can be very deleterious, programmed DSBs are induced and repaired in a highly regulated manner during meiosis to ensure interhomolog CO formation, which is critical for achieving accurate chromosome segregation at meiosis I (*Martinez-Perez and Colaiácovo, 2009*). Regulatory mechanisms need to be set in place to ensure that sufficient DSBs are formed to generate at least one CO per pair of homologous chromosomes and then turn off DSB formation once interhomolog engagement is achieved. Studies in different model organisms have described the existence of feedback mechanisms to regulate meiotic DSB number and distribution.

Work done in both mice and Drosophila has shown that ATM kinase, which is activated in response to DNA damage, is also activated by SPO11-mediated DSBs and is implicated in controlling meiotic DSB levels by inhibiting further DSB formation via a negative feedback mechanism (*Lange et al., 2011*; *Joyce et al., 2011*). Studies in yeast, have shown both *cis-* and *trans*-acting regulation of DSB formation acting respectively either along the same or between different homologous chromosomes or sister chromatids. During regulation in *cis*, presence of a strong DSB hotspot inhibits DSB formation nearby (*Wu and Lichten, 1995*; *Xu and Kleckner, 1995*; *Fukuda et al., 2008*) whereas during regulation in *trans*, DSB formation at one site inhibits DSB formation at either the same site or a nearby site on the homologous partner. Communication between interhomolog interaction and DSB formation is important to establish an even spacing of total recombination events, and the Mec1 (ATR) DNA damage response kinase is proposed to mediate this by *trans* inhibition (*Zhang et al., 2011*). Meanwhile, the Tel1 (ATM) kinase exerts *cis* inhibition at the local scale, on the same chromatid and between sister chromatids (*Garcia et al., 2015*). Following meiotic DSB formation, Tel1 and Mec1 may phosphorylate Rec114, a Spo11-accessory protein required for DSB formation, which has been proposed to inhibit further DSB formation by reducing the interaction of Rec114 with DSB hotspots (*Carballo et al., 2013*). In *C. elegans*, our data suggests that PLK-1/2-dependent phosphorylation of SYP-4 functions as a signal to constrain further DSB formation when one of the DSBs undergoes CO designation/precursor formation. However, we cannot rule out a role for ATM/ATR kinases in PLK-1/2-mediated DSB inhibition.

Studies in yeast and mice suggest that defects in interhomolog engagement result in increased DSB formation. In yeast, the ZMM proteins (Zip1-4, Msh4-5, Mer3, and others) are required for both SC and CO formation and studies in *zmm* mutants implicate interhomolog engagement as directly responsible for inhibiting DSB formation (*Keeney et al., 2014*). In mice, unsynapsed chromosomes exhibit elevated DSB levels (*Kauppi et al., 2013*; *Thacker et al., 2014*; *Hayashi et al., 2010*; *Lam and Keeney, 2015*) and the illegitimate (non-homologous) synapsis observed in *Spo11*[-/-] mutants is sufficient to evict the HORMA-domain proteins HORMAD1/2 from chromosomes (*Wojtasz et al., 2009*) suggesting a role for SC formation in the feedback mechanism that inhibits further DSB formation once interhomolog engagement is achieved. However, the molecular basis for transmission of the interhomolog engagement-mediated feedback loop on DSB formation is not understood. Axial element proteins Hop1 and Red1 in yeast, HTP-1 and HTP-3 in *C. elegans*, and HORMAD1, which is a component of unsynapsed axes in mice, have been shown to be required for normal levels of programmed DSBs (*Mao-Draayer et al., 1996*; *Woltering et al., 2000*; *Blat et al., 2002*; *Niu et al., 2005*; *Carballo et al., 2008*; *Lam and Keeney, 2015*; *Goodyer et al., 2008*; *Couteau and Zetka, 2005*; *Martinez-Perez and Villeneuve, 2005*; *Sanchez-Moran et al., 2007*;

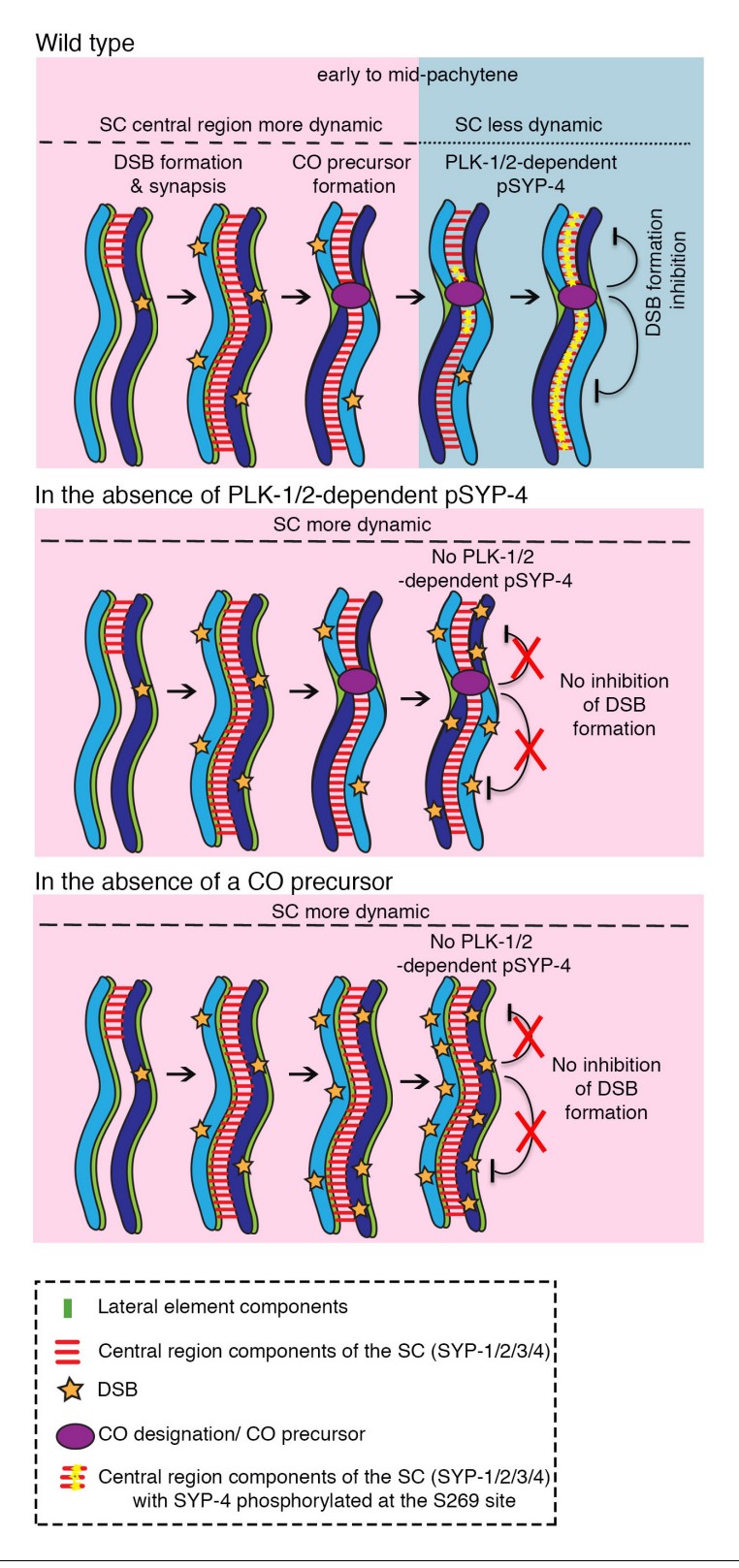

**Figure 7.** Model for how PLK-1/2-dependent phosphorylation of the synaptonemal complex protein SYP-4 regulates double-strand break formation through a negative feedback loop. In wild type, PLK-1/2-dependent phosphorylation of SYP-4 at the S269 site occurs in response to CO designation. This phosphorylation switches the central region of the SC from a more dynamic (pink) to a less dynamic state (blue) during pachytene, inhibiting

*Figure 7 continued on next page*

*Figure 7 continued*

additional DSB formation on the homologous chromosomes. In the phosphodead mutant, in the absence of phosphorylation of SYP-4 at the S269 site, the SC fails to stabilize and the absence of a feedback loop results in continued DSB formation along the homologous chromosomes. Our model suggests that in the absence of CO designation, PLK-1/2-dependent phosphorylation of SYP-4 does not take place and the SC persists in a more dynamic state, which allows the homologs to remain in a DSB-permissive state.

---

*Latypov et al., 2010*). Our work implicates a central region component of the SC in functioning as a signaling unit to inhibit further DSB formation along synapsed chromosomes once a CO is designated or after CO precursor formation in *C. elegans*. This SC-mediated feedback mechanism may also be present in humans since a central element protein of the SC, SYCE1, has PLK phosphorylation sites at S29 and S191 predicted by the phosphorylation site predictor GSP polo and both of these sites are identified as phosphorylated in vivo in humans by Phosphosite Plus (*Hornbeck et al., 2015*).

## Materials and methods

### *C. elegans* strains and genetics

*C. elegans* strains were cultured at 20°C under standard conditions and the N2 Bristol strain was used as the wild-type background (*Sulston and Brenner, 1974*). The following mutations and chromosome rearrangements were used: **LG I:** *syp-4(rj48 (S269A) I, syp-4(rj49 (S269D) I, syp-4(rj48 (S269A) I; meIs9(unc-119(+) pie-1promoter::gfp::syp-3, syp-4(rj49 (S269D) I; meIs9(unc-119(+) pie-1promoter::gfp::syp-3, syp-4(rj48 (S269A) I; meIs8 [pie-1p::GFP::cosa-1 + unc-119(+)] II, syp-4(rj49 (S269D) I; meIs8 [pie-1p::GFP::cosa-1 + unc-119(+)] II, syp-4(tm2713) I/hT2 [bli-4(e937) let-?(q782) qIs48] (I;III), plk-2(ok1936) I, zhp-3(jf61)/hT2[bli-4(e937) let-?(q782) qIs48 (I;III)]I.* **LG II:** *rol-1(e91) dsb-2 (me96)/mnC1 (dyp-1(e128) unc-52(e444)) II.* **LG III:** *cosa-1(me13)/qC1(qIs26) III.* **LG IV:** *dsb-1(tm5034) IV/nT1(unc-?(n754) let-?) (IV;V), syp-1(me17) V/nT1 [unc-?(n754) let-? qIs50] (IV;V), spo-11(ok79) IV/ nT1 [unc-?(n754) let-?] (IV;V), mre-11(ok179) IV/nT1 [unc-?(n754) let-?] (IV;V), rad-51(lg8701) IV/nT1 [let-?(m435)] (IV;V), rad-54 (tm1268)/ hT2 [bli-4(e937) let-?(q782) qIs48] (I;III).*

### Antibodies

A rabbit phospho-specific polyclonal antibody was generated by Abmart (China) using the phospho-peptide C-QFDR(pS)FILAS encompassing the S269 site in SYP-4. Two peptides were synthesized: An antigen peptide with phosphoserine C-QFDR(pS)FILAS and a control peptide without phosphorylation (C-QFDRSFILAS). Serum harvested from the rabbits immunized with the antigen peptides went through two rounds of affinity purification. First, serum was passed through a column to which the antigenic peptide was coupled in order to isolate phospho-specific SYP-4 antibodies. The eluate from the first column was then passed through a second column to which the control peptide was coupled and flow through was collected. This second step was done to remove any non-phosphorylated SYP-4 antibodies. An ELISA titer of $\geq$1:50,000 against the modified peptide and a modified/ unmodified titer ratio of $\geq$8 were used as validation criteria.

A rabbit polyclonal antibody was generated against a SYP-2 peptide (RRVSFASPVSSSQ) by Abmart and used at a 1:200 dilution for immunofluorescence. A polyclonal antibody against a SYP-1 peptide (VDAPTEALIETPVDDQSSGFLC) was generated by the GenScript Antibody Group (Piscataway, NJ) and used at a 1:3000 dilution for immunofluorescence and 1:5000 dilution for western blot analysis.

All other primary antibodies were used at the following dilutions for immunofluorescence: chicken α-GFP (1:400; Abcam, Cambridge, MA), rabbit α-SYP-1 (1:200; *MacQueen et al., 2002*), rabbit α-SYP-3 (1:200; *Smolikov et al., 2007b*), rabbit α-SYP-4 (1:200; *Smolikov et al., 2009*), guinea pig α-HTP-3 (1:400; *Goodyer et al., 2008*), rabbit α-RAD-51 (1:10,000; Novus Biological (SDI), Littleton, CO), rabbit α-HIM-8 (1:500; Novus Biological (SDI)), rabbit α-PLK-1 (1:50; *Labella et al., 2011*), rabbit α-PLK-2 (1:200; *Nishi et al., 2008*), guinea pig α-SUN-1 Ser8-pi (1:700; *Woglar et al., 2013*). The following secondary antibodies from Jackson ImmunoResearch (West Grove, PA) were used at a

1:200 dilution: α-chicken FITC, α-rabbit Cy3, α-goat alexa 488, and α-guinea pig alexa 488. Vecta-shield containing 1 µg/µl of DAPI from Vector Laboratories (Burlingame, CA) was used as a mounting media and anti-fading agent.

Primary antibodies were used at the following dilutions for western blot analysis: rabbit α-SYP-3 (1:200; *Smolikov et al., 2007b*), mouse α-tubulin (1:2000; Sigma, St. Louis, MO), and histone H3 (1:2000; Abcam). HRP-conjugated secondary antibodies, donkey anti-goat, rabbit anti-mouse, and mouse anti-rabbit from Jackson ImmunoResearch were used at a 1:10,000 dilution.

## Immunofluorescence and imaging

Whole mount preparation of dissected gonads and immunostaining procedures were performed as in (*Colaiácovo et al., 2003*). Immunofluorescence images were captured with an IX-70 microscope (Olympus, Waltham, MA) fitted with a cooled CCD camera (CH350; Roper Scientific) driven by the Delta Vision system (Applied Precision, Pittsburgh, PA). Images were subjected to deconvolution by using the SoftWoRx 3.3.6 software (Applied Precision).

## Generation of mutants via the CRISPR-Cas9 system

CRISPR-Cas9 genome editing technology was used to engineer *syp-4* phosphodead and phosphomimetic mutations at the endogenous locus (*Tzur et al., 2013*). To generate a phosphodead *syp-4* mutant, serine 269 (S269A) was mutated to alanine. To generate a phosphomimetic mutant, serine 269 (S269D) was mutated to aspartic acid. We used the sgRNA recognition site (AATTTGTGGAAG TCTCAGTCTGG) 2414 base pairs downstream of the start codon. We cloned the sgRNA in to the *pU6::unc-119*_sgRNA plasmid (*Friedland et al., 2013*). The donor sequence containing the genomic sequence of *syp-4* extending from 943 bp upstream to 2128 bp downstream of the start codon with the following changes: TC to GA change at positions 1021 and 1022, to generate the phosphomimetic mutant, and a T to G change at position 1021, to generate the phosphodead mutant (resulting in a silent mutation that is expected to prevent re-cutting by Cas9), was cloned into the BglII site of the pCFJ104 vector expressing *Pmyo-3::mCherry::unc-54* (*Frøkjaer-Jensen et al., 2008*). A cocktail consisting of a plasmid expressing the sgRNA (200 ng/µl), a plasmid expressing the donor sequence (97.5 ng/µl), Cas9 (200 ng/µl) and the co-injection marker pCFJ 90 (*Pmyo-2::mCherry::unc-54utr*; 2.5 ng/µl) was microinjected into the gonad arms of the worms (P0s). F1 animals expressing the co-injection marker were sequenced to identify mutants and homozygous animals were picked from among the F2 generation and confirmed by sequencing.

## Irradiation experiments

Young (~18 hr post-L4 stage) wild type and *spo-11* mutant adult animals were irradiated with approximately 10 Gy from a $Cs^{137}$ source. Irradiated and untreated control worms were dissected 6–8 hr post-irradiation and immunostained with the phospho-specific SYP-4 antibody.

## RNA interference

Feeding RNAi experiments were performed as in (*Govindan et al., 2006*) with the following modifications: three L4-stage animals were placed on each RNAi plate and 24 hr post-L4 animals from the next generation were screened for a phenotype. Control RNAi was performed by feeding HT115 bacteria expressing the empty pL4440 vector. For complete depletion of *rad-54*, L1-stage animals were placed on the RNAi plates and F2 animals were scored for a phenotype. Animals were grown on the *rad-54(RNAi)* plates the entire time.

## Fluorescence in situ hybridization (FISH)

FISH probes were generated as in (*Smolikov et al., 2007b*). The 5S rDNA probe was generated by PCR with the primers 5′-TACTTGGATCGGAGACGGCC-3′ and 5′-CTAACTGGACTCAACGTTGC-3′. FISH probes were labeled using Terminal Transferase (NEB M0315S, Ipswich, MA) with Fluorescein-12-dCTP (Perkin Elmer NEL-424, Waltham, MA). The average numbers of nuclei scored per zone (n) for a given genotype to analyze pairing for chromosome V using FISH and pairing for the X chromosome using HIM-8 staining were as follows: zone 1 (n = 263), zone 2 (n = 145), zone 3 (n = 157), zone 4 (n = 153), zone 5 (n = 148), and zone 6 (n = 134). Statistical comparisons between genotypes were conducted using the two-tailed Mann-Whitney test, 95% confidence interval (C.I.).

### RAD-51 time course analysis

Quantitative analysis of RAD-51 foci/nucleus was performed as in (*Colaiácovo et al., 2003*). The average number of nuclei scored per zone (n) for a given genotype was as follows: zone 1 (n = 235), zone 2 (n = 122), zone 3 (n = 119), zone 4 (n = 108), zone 5 (n = 99), and zone 6 (n = 95). Statistical comparisons between genotypes were conducted using the two-tailed Mann-Whitney test, 95% C.I.

### Germ cell apoptosis

Germ cell corpses were scored in adult hermaphrodites (~18 hr post-L4) as in (*Kelly et al., 2000*). Statistical comparisons between different genotypes were conducted using the two-tailed Mann-Whitney test, 95% C.I.

### FRAP image acquisition

Day-1 (~18 hr post-L4) animals expressing SYP-3::GFP were anaesthetized with 0.1% levamisole and loaded with levamisole onto 4% agarose pads on slides (VWR; cat#16004–368, Radnor, PA) and then covered with No. 1.5 coverslips. Images were acquired with a Nikon 60X/1.4 Plan Apo VC objective on an inverted Nikon Ti Microscope with Perfect Focus System and a Spectral Borealis-modified Yokogawa CSU-X1 spinning disc confocal head. EGFP fluorescence was excited with a 488 nm solid-state laser (~220µW at the sample) and emission was collected through an ET525/50m emission filter (Chroma, Bellows Falls, VT) onto an ORCA-ER CCD camera (Hamamatsu, Boston, MA) with binning of $2 \times 2$ for an effective pixel size of 217 nm. Z-stacks of five planes with a 0.75 µm step-size were acquired every 30 s for a total of 80 volumes (40 min). After two baseline volumes were acquired, a diffraction-limited volume in roughly 10 nuclei across the field of view was photobleached throughout the Z-stack encompassing the entire depth of the nuclei using a MicroPoint (Photonic Instruments, Saint Charles, IL) equipped with a Coumarin 440 dye cell; MicroPoint laser power was adjusted in pilot experiments so that bleaching resulted in a ~90% reduction in local fluorescence intensity. Hardware and image acquisition were controlled with MetaMorph 7.8.4 (Molecular Devices, Sunnyvale, CA).

### FRAP analysis

Z-stack sum-intensity projections were used for FRAP analysis. Images were qualitatively evaluated for animal health, and data sets in which animals ceased movement during the time-series were rejected. Gross lateral animal movement and drift were corrected for translation and rotation using the StackReg plugin (*Thévenaz et al., 1998*) in Fiji (*Schindelin et al., 2012*), rectangular regions encompassing individual nuclei were extracted for further rigid body registration in StackReg to minimize the impact of chromosomal movement on intensity measurements. Only nuclei for which the photobleached region (region of interest, ROI) was in focus for the entire time period of fluorescence recovery were examined, whereas nuclei demonstrating obvious rotation around the lateral (XY) plane were excluded from analysis. After registration, the average intensity of the bleached region in each nucleus was measured over the duration of the time-lapse by manually tracing the photobleached region to correct for chromosomal movement. Background measured in a non-fluorescent region of the image was subtracted from each time point, and intensity measured in a region of interest encompassing all nuclei was used for double-normalization to correct for photobleaching during acquisition as well as lost signal due to the bleach pulse as described in (*Phair et al., 2004*). Double-normalized traces were fit to a one-phase association curve: *Y=A\*(1-e-Kt)*, where *A* is the mobile fraction (fit plateau), and *K* is the association rate. Traces that could not be fit to a one-phase association were rejected. The significance of the differences between the mobile fractions of different groups was assessed with a Kruskal-Wallis test. Statistical analysis was performed in Prism six (GraphPad, La Jolla, CA). The difference across groups was $p < 0.0001$.

## Acknowledgements

We are grateful to the Caenorhabditis Genetics Center for providing strains, to M. Zetka for the HTP-3 and PLK-1 antibodies, R. Lin for the PLK-2 antibody, and V. Jantsch for the SUN-1 S8pi antibody. We thank J. Engebrecht and members of the Colaiácovo lab for critical reading of this manuscript and for providing helpful suggestions. We thank Hunter Elliott in the Image and Data Analysis

Core at Harvard Medical School for advice on FRAP image analysis and the Nikon Imaging Center at Harvard Medical School for use of their microscopes. This work was supported by National Institutes of Health grant R01GM072551 to MPC and a fellowship from the Lalor Foundation to SN.

## Additional information

### Funding

| Funder | Grant reference number | Author |
|---|---|---|
| National Institutes of Health | R01GM072551 | Monica P Colaiácovo |
| Lalor Foundation | | Saravanapriah Nadarajan |

The funders had no role in study design, data collection and interpretation, or the decision to submit the work for publication.

### Author contributions

SN, Conceptualization, Formal analysis, Validation, Investigation, Visualization, Methodology, Writing—original draft, Writing—review and editing; TJL, Formal analysis, Visualization, Methodology, Writing—original draft; EA, JG, Formal analysis, Investigation, Writing—review and editing; MDB, Resources, Formal analysis, Writing—review and editing; JCW, Resources, Supervision, Visualization, Writing—review and editing; MPC, Conceptualization, Supervision, Funding acquisition, Writing—original draft, Project administration, Writing—review and editing

### Author ORCIDs

Monica P Colaiácovo, http://orcid.org/0000-0001-7803-4372

## Additional files

### Supplementary files

• Supplementary file 1. Brood size, embryonic lethality and incidence of males observed for *syp-4 (S269A)* and *syp-4(S269D)* mutants. The 'Eggs Laid' column indicates the average number of eggs laid (including both non-hatched and hatched embryos) per P0 hermaphrodite ± standard deviation. % Embryonic lethality was calculated by dividing the number of non-hatched embryos by the total number of hatched and non-hatched embryos laid. % Males was calculated by dividing the total number of males observed by the total number of hatched (viable) progeny scored. N = total number of P0 worms for which entire broods were scored. *p<0.0001 (Two-tailed Mann-Whitney test, 95% C.I.).

• Supplementary file 2. Dunn's multiple comparisons test of FRAP analysis. Dunn's multiple comparisons test between groups was applied to assess statistical significance of GFP::SYP-3 fluorescence recovery data presented in *Figure 6B*. LZ indicates leptotene/zygotene stage nuclei and MP indicates mid-pachytene stage nuclei.

• Supplementary file 3. Numerical data for brood size, embryonic lethality and incidence of males shown on *Supplementary file 1*.

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
