## [Decision Letter]

Thank you for submitting your article "Polo-like kinase-dependent phosphorylation of SYP-4 regulates double-strand break formation via a negative feedback loop" for consideration by *eLife*. Your article has been reviewed by three peer reviewers, and the evaluation has been overseen by a Reviewing Editor (Scott Keeney) and Jessica Tyler as the Senior Editor. The reviewers have opted to remain anonymous.

The reviewers have discussed the reviews with one another and the Reviewing Editor has drafted this decision to help you prepare a revised submission.

Summary:

The manuscript by Nadarajan et al. addresses an important question of the interplay between synapsis and recombination, particularly how PLK-2 phosphorylation of SYP-4 regulates recombination by affecting the ability to continue to form DSBs.

Recent work has revealed the existence of mechanisms that ensure that meiotic DSB formation is shut off when certain conditions along homologous chromosomes are met. In addition, signal transduction pathway(s) that limit DSBs have been identified. In a number of organisms, including yeast, mouse and *Drosophila*, the ATM kinase signal transduction pathway limits DSBs, both along the genome and locally. Moreover, assembly of the SC central element is implicated in limiting DSB formation, as suggested by increased DSB formation in budding yeast mutants that abrogate SC assembly. In general, however, these regulatory mechanisms remain poorly understood.

This manuscript describes the function of PLK-2 dependent phosphorylation of SYP-4 in *C. elegans*. Phosphorylation of SYP-4 depends on double strand break formation (DSB), correlates with the initiation of meiosis specific crossover recombination and is required for wildtype levels of DSB formation. Evidence is presented that this phosphorylation event controls the dynamics of the synaptonemal complex: In early meiotic prophase, the SC is more dynamic, incorporating new components of the SC, such as SYP-3, more readily. This dynamicity is lost later in meiotic prophase and correlates with SYP-4 phosphorylation. When the SYP-4 phosphoacceptor site is mutated to alanine, preventing phosphorylation, the SC is dynamic longer than in wildtype worms and when the SYP-4 phosphoacceptor site is mutated to aspartic acid, mimicking constitutive phosphorylation, the SC is less dynamic earlier than in wildtype worms. The authors present a model in which the phosphorylation of SYP-4 in a PLK dependent manner occurs in response to the formation of a crossover precursor, inhibiting the formation of additional DSBs and promoting the stability of the SC.

The authors' experiments support most of the conclusions of the paper and the model they propose. The manuscript is clearly written and provides important information about a crucial mechanism that links the stability of synaptonemal complex, the progression of meiotic recombination and the regulation of DSB formation.

Essential revisions:

The reviewers and reviewing editor raised a number of concerns that must be adequately addressed before the paper can be accepted. Some of the required revisions will likely require further experimentation within the framework of the presented studies and techniques.

Additional experimental data needed:

1) More evidence/validation for phospho-specific antibody is needed, since this is such a critical tool. For example, structural changes may be happening that might affect e.g. epitope accessibility. Does the anti-pSYP-4 work for western? If so, show a western of anti-pSYP-4 and sensitivity of the band to phosphatase treatment. At a bare minimum, the authors need to show that SYP-4 is still recruited along chromosomes under conditions where the presumed phospho-specific SYP-4 antibody does not detect a signal, especially in the *plk* mutant condition. The principal evidence for SYP-4-phopho-specificity of the antibody is that the signal is delayed or disappears in the plk-2 and/or plk-1 mutant. But in the same mutant, SYP-1 assembly is also compromised, at a similar stage (e.g. Harper et al., 2011, Figure 1). The interdependence between the SYPs was shown in null mutants, but *syp-4(S269A)* could be functional for SC assembly but unstable/reduced later assuming it is not required for SC maintenance. Details should be given about the affinity purification of the antibody: Was that done with a S269-nonphospho-peptide? How was success of the affinity-purification determined?

2) Please document the effect of *syp-4* point mutants on chromosome segregation or offspring viability (apart from the mild increase in apoptosis).

3) Quantify and document better the per-chromosome correlation of COSA-1 and pSYP-4. From Figure 3, it doesn't seem that stretches of pSYP 4 signal are first observed only on chromosomes that have a GFP::COSA-1 foci. Some quantification as well as better images should be provided. For example, one could determine the contour of the stretch of pSYP-4 staining and calculate the association of stretches with the COSA-1 foci.

4) Define single pSYP-4 chromosome in more detail. What is the cause for the single chromosome staining: Is there a second wave of PLK-1 phosphorylation? Is it known which chromosome track stains with pSYP-4 in the DSB break processing mutants? Is it always the same chromosome track? Can some FISH be done to address this? This is off-topic of the main point of the paragraph but it suggests that some chromosome stretches get pro-crossover factors which pSYP-4 can respond to. However, is that a reasonable assumption? This is somewhat important in the interpretation of the pairing data that is presented subsequently. Is there no difference in the pairing data because one chromosome behaves aberrantly from the rest and that happens to be the chromosome you which you tested for pairing? It also would be nice to have some speculation statement about the chromosome track.

5) Address concerns about chromosome movements within cells for FRAP experiments (subsection “FRAP Analysis”) in regards to FRAP analysis in Figure 6. Although the impact of cellular movement was minimized for intensity measurements, what about chromosomal movements within the cell? Meiotic prophase generally has been shown to display significant changes in motion of its chromosomes. It would be important to have data and a statement that this was not a concern in generating the change of fluorescent intensity. This is particularly important because z-projections were used for the analysis. Chromosome motion instead of incorporation of central element may or may not be distinguished well in projections.

Major points that can be addressed by text fixes or changes to figures with available data:

6) Incorporate SC dynamics (and how they change) into the Figure 7 model. It is also suggested the authors should show the proposed feed-back loop, i.e. showing that DSBs trigger an event that in turn triggers their own switch off.

7) The authors are careful to point out in the Results section that their data are consistent with phosphorylation of SYP-3 being PLK dependent. However, they start their Discussion with: "We have discovered that phosphorylation of a central region component of the SC, SYP-4, by Polo-like kinases…" and "Our data indicate that Polo-like kinases phosphorylate SYP-4 starting at early pachytene…" The authors haven't provided any data demonstrating that PLK-1 or PLK-2 directly phosphorylates SYP-4. They can only say the phosphorylation is dependent on PLK-1/2 and that their data suggests that polo-like kinases phosphorylate SYP-4 in early pachytene, consistent with PLK-2's localization.

8) In Smolikov et al., PLOS Genetics (2009), elevated levels of persistent RAD-51 foci were interpreted as a defect in recombination progression, rather than as a failure to shut-off DSBs. Given the SYP-4 involvement in DSB shut-off, this might be a good opportunity to reevaluate the earlier findings. More generally, this would also be a good opportunity to reevaluate the implications of using rad-54 or other recombination-defective mutations as a tool to estimate total DSB levels in wild type.

9) Provide better introduction and discussion of current results relative to other organisms. Please introduce and discuss similarities and differences of DSB limitation via Plk in *C. elegans* compared to the role of ATM/Tel1 in other organisms. In addition, the discussion of similarities to the homolog engagement pathway for DSB regulation that has been previously documented in yeast and mouse needs to be improved. In the Introduction it is stated: "However, once a DSB is designated to become a CO a feedback mechanism turns off further programmed DSBs from forming (Lam and Keeney 2015). The molecular basis for transmission of this feedback regulation remains an open question." The first statement is not a fair summary of what is known. And the implication from second sentence seems to be that the current results will address the lack of information, but the connection of the current findings to these other organisms is not convincing. CO designation is one possibility for why "homolog engagement" turns off DSB formation in yeast and mouse, but it is not the only possibility. Keeney's and Toth's labs propose that it is SC formation that triggers DSB inhibition, in part because even non-homologous SC formation in the absence of SPO11 is accompanied by TRIP13-dependent HORMAD depletion from chromosomes in mouse (see Wojtasz 2009, Thacker 2014, and Keeney 2014). In any case, the text shouldn't state it so categorically that DSB inhibition is a consequence of CO designation, because there is certainly no evidence to strongly support that conclusion. Concerning the molecular basis of the feedback: indeed this is not known. But it clearly does not require polo kinase in yeast (Cdc5) because Keeney's lab showed that the homolog engagement (ZMM-dependent) pathway operates even in an ndt80 mutant (Thacker 2014). Cdc5 is not expressed in the absence of Ndt80. Minor point: the Lam and Keeney 2015 paper does not seem to be a particularly apt reference for this general point. Perhaps Keeney et al. 2014 would be better.

10) Subsection “PLK-2 localization on chromosomes during pachytene is dependent on the SYP proteins”. In Figure 1, the localization in PLK-2 to the nuclear periphery of syp-4 is qualitatively different than in WT. In fact, PLK-2 location is different between *syp-1* and *syp-4* mutants. This is in contrast to the statement in the figure legend that PLK-2 localization is not different. So, either the figure for the *syp-4* mutant is not representative or there is a difference that exists. There is also a distinct difference in the HTP-3 staining for *syp-4*. Some comment should be made.

11) In Figure 1, the image contrast/ detection thresholds for WT and *syp* mutants appear to be set differently. If anything, the background should be set lighter in *syp* mutants. The same concern applies for PLK-1 in Figure 1—figure supplement.1B, and Figure 2 the *syp-4* and *plk-2* panels. In Figure 3, the colored boxes are hard to see and the fact that different antibodies and stages are identically color-coded is confusing (e.g. green stands for COSA-1 and also midpachytene).

12) Primary data should be provided for RAD-51 foci in the S269A mutant (i.e. a representative IF image).

13) The possible cause/effect relationship between SC dynamics and extra RAD-51 should be discussed clearly. While the inference is that continued SC dynamics result in extra DSBs, currently there is no evidence to exclude the reverse relationship. If additional experimental data are available, great; if not, adding discussion would suffice. For example, have the authors explored whether the SC in *syp-4* (A269S) still exhibits high dynamics in the absence of DSBs/SPO-11? If *syp-4* (S269A)-induced SC dynamics persist despite absence of DSBs, this would allow excluding a role of extra DSBs in inducing SC dynamic. Or: does the S269D phosphomimetic mutation eliminate SC dynamics in absence of DSBs? Notably, even in absence of DSBs, some SYP-4 phosphorylation is still happening. Is it possible to compare the SC dynamics phenotypes of S269A and other recombination mutant(s) that assemble apparently normal SC?

14) Does SYP-4 phos work via chromosome association of DSB-1 and -2? It would be interesting to discuss this possibility.

15) It seems that the authors' model SYP-4 phos should be more important when total DSBs are low (e.g., in a *dsb-2* mutant or *spo-11* + low IR dose). Did the authors test this? If not, it could be a Discussion point.

---

## [Author Response]

*Essential revisions:*

*The reviewers and reviewing editor raised a number of concerns that must be adequately addressed before the paper can be accepted. Some of the required revisions will likely require further experimentation within the framework of the presented studies and techniques.*

*Additional experimental data needed:*

*1) More evidence/validation for phospho-specific antibody is needed, since this is such a critical tool. For example, structural changes may be happening that might affect e.g. epitope accessibility. Does the anti-pSYP-4 work for western? If so, show a western of anti-pSYP-4 and sensitivity of the band to phosphatase treatment.*

Unfortunately, the anti-pSYP-4 antibody does not work on western blots.

*At a bare minimum, the authors need to show that SYP-4 is still recruited along chromosomes under conditions where the presumed phospho-specific SYP-4 antibody does not detect a signal, especially in the plk mutant condition. The principal evidence for SYP-4-phopho-specificity of the antibody is that the signal is delayed or disappears in the plk-2 and/or plk-1 mutant. But in the same mutant, SYP-1 assembly is also compromised, at a similar stage (e.g. Harper et al., 2011, Figure 1). The interdependence between the SYPs was shown in null mutants, but syp-4(S269A) could be functional for SC assembly but unstable/reduced later assuming it is not required for SC maintenance.*

We now provide several lines of evidence that the pSYP-4 antibody is specific. First, in Figure 6—figure supplement 1, we show that SYP-4 is recruited along the homologous chromosomes in the *syp-4(S269A)* phosphodead mutant but we do not detect phosphorylated SYP-4 signal in this mutant background (Figure 2), which argues that the pSYP-4 signal is specific and not a consequence of SYP-4 not being loaded or maintained on the chromosomes. Second, as the reviewers suggested, we now show SYP-4 in addition to SYP-1 localization in the *plk-2;plk- 1(RNAi)* mutant (Figure 2—figure supplement 1 and Figure 2). This new data is now mentioned as follows: “We found that pSYP-4 signal, but not SYP-4, was completely lost upon depletion of *plk-1* by RNAi in the *plk-2* mutant (Figure 2 and Figure 2—figure supplement 1)”. In the *plk-2;plk-1(RNAi)* mutant, we observe an uneven localization for both SYP-1 and SYP-4 along the homologous chromosomes. Specifically, SYP-1 and SYP-4 form tracks of uneven intensity along DAPI-stained chromosomes, and we now clearly indicate that in our revised figure legends for both Figure 2 and Figure 2—figure supplement 1. However, even where we observe strong signal for SYP-1 or SYP-4 we still do not observe pSYP-4 signal. We would like to clarify that we did not argue that SYP protein localization is normal in this mutant background. Rather, even though the SYP proteins are able to localize, albeit unevenly, on the chromosomes we did not detect any SYP-4 phosphorylation on those chromosomes. Third, in the recombination defective mutants, even though SYP-1 can still load on the chromosomes, we do not see pSYP-4 localization, except for a single chromosome track in 19.7% of nuclei in mutants for factors involved in early steps of recombination (the pSYP-4 signal is completely absent in pro-crossover mutants) (Figure 3 and Figure 3—figure supplement 1). Finally, we do not observe any obvious defects in SC maintenance in the *syp-4(S269A)* mutant (Figure 6—figure supplement 1). Together, these data strongly support the specificity of the phospho SYP-4 antibody.

*Details should be given about the affinity purification of the antibody: Was that done with a S269-nonphospho-peptide? How was success of the affinity-purification determined?*

As per the reviewer’s request, we now provide additional information about the affinity purification method used for the SYP-4 phospho-specific antibody in the Materials and methods subsection “Antibodies”. A rabbit phospho-specific polyclonal antibody was generated by Abmart using the phospho-peptide C-QFDR(pS)FILAS encompassing the S269 site in SYP-4. Two peptides were synthesized: An antigen peptide with phosphoserine C-QFDR(pS)FILAS and a control peptide without phosphorylation (C-QFDRSFILAS). Serum harvested from the rabbits immunized with the antigen peptides went through two rounds of affinity purification. First, serum was passed through a column to which the antigenic peptide was coupled in order to isolate phospho- specific SYP-4 antibodies. The eluate from the first column was then passed through a second column to which the control peptide was coupled and flow through was collected. This second step was done to remove any non-phosphorylated SYP-4 antibodies. An ELISA titer of ≥ 1:50,000 against the modified peptide and a modified/unmodified titer ratio of ≥ 8 were used as validation criteria. In addition, phospho- specificity of the antibody was tested by immunofluorescence in *syp- 4(S269A)* phosphodead mutants (Figure 2).

*2) Please document the effect of syp-4 point mutants on chromosome segregation or offspring viability (apart from the mild increase in apoptosis).*

We have now added [Supplementary-material SD7-data] reporting the brood size, percentage of embryonic lethality, and percentage of males observed in the *syp- 4* point mutants compared to wild type. This new data is now mentioned as follows: “We observed a significant reduction in brood size in both *syp- 4(S269A)* phosphodead and *syp-4(S269D)* phosphomimetic mutants compared to wild type ([Supplementary-material SD7-data]). However, we only saw 6-8% embryonic lethality accompanied by 0.3-1% males among the progeny from these mutants, suggesting only a mild increase of autosomal and X chromosome nondisjunction during meiosis in these mutants, which could stem in part from the <4% of oocytes observed having a reduced number of COSA-1 foci ([Supplementary-material SD7-data]).”

*3) Quantify and document better the per-chromosome correlation of COSA-1 and pSYP-4. From Figure 3, it doesn't seem that stretches of pSYP 4 signal are first observed only on chromosomes that have a GFP::COSA-1 foci. Some quantification as well as better images should be provided. For example, one could determine the contour of the stretch of pSYP-4 staining and calculate the association of stretches with the COSA-1 foci.*

As per the reviewer’s suggestion, we now show the quantification of the per- chromosome correlation of COSA-1::foci and pSYP-4 in Figure 3—figure supplement 1. We also included high-magnification images in Figure 3 to show the colocalization of pSYP-4 with COSA-1::GFP in early, mid and late pachytene nuclei. The new data is now mentioned as follows: “Strikingly, at that same stage, phosphorylated SYP-4 was first observed on 76% of chromosomes (n=78) with GFP::COSA-1 foci, whereas a low percentage of chromosomes showed either only pSYP-4 signal (5.8%; n=6) or COSA-1 foci (18.4%; n=19) (Figure 3 and Figure 3—figure supplement 1)”.

*4) Define single pSYP-4 chromosome in more detail. What is the cause for the single chromosome staining: Is there a second wave of PLK-1 phosphorylation? Is it known which chromosome track stains with pSYP-4 in the DSB break processing mutants? Is it always the same chromosome track? Can some FISH be done to address this? This is off-topic of the main point of the paragraph but it suggests that some chromosome stretches get pro-crossover factors which pSYP-4 can respond to. However, is that a reasonable assumption? This is somewhat important in the interpretation of the pairing data that is presented subsequently. Is there no difference in the pairing data because one chromosome behaves aberrantly from the rest and that happens to be the chromosome you which you tested for pairing? It also would be nice to have some speculation statement about the chromosome track.*

To address the reviewer’s questions we now show that the COSA-1::GFP foci are observed on the chromosomes with the pSYP-4 signal in *spo-11* mutants (Figure 3—figure supplement 3) suggesting that SYP-4 phosphorylation occurs in response to CO precursor formation, which is consistent with our model. We also examined which chromosome exhibited the pSYP-4 track in *spo-11* mutants. Specifically, we performed HIM-8 staining to mark the X chromosome and FISH to identify chromosomes III and V in the *spo-11* mutant. We found that 46% of pSYP-4 tracks corresponded to the X chromosome, 11.6% to chromosome V and 9.5% to chromosome III. The higher frequency of pSYP-4 signal along the X chromosome could be due to its distinct chromatin structure and/or a result of it undergoing delayed replication compared to autosomes (Bender et al., 2004; Kelly et al., 2002; Gao et al., 2015; Jaramillo-Lambert et al., 2007), which might result in DSBs being carried into meiosis. We now include all this new data as Figure 3—figure supplement 3, and in the main text as follows: “Moreover, a COSA-1::GFP focus was observed on each of these chromosomes exhibiting pSYP-4 signal (Figure 3—figure supplement 3). […] The higher frequency of pSYP-4 tracks along the X chromosome could be due to its distinct chromatin structure and/or a result of it undergoing delayed replication compared to autosomes (Bender et al., 2004; Kelly et al., 2002; Jaramillo-Lambert et al., 2007), which might lead to a higher number of DNA lesions that could be processed for CO precursor formation during meiosis.”

Regarding whether there may be a second wave of PLK-1 phosphorylation that could result in this single chromosome with pSYP-4 staining, we were not able to detect PLK-1 signal on chromosomes during pachytene in either wild type or *plk-2* mutants (Figure 1—figure supplement 1), although we cannot rule out the possibility that there is some PLK-1 that is below threshold levels of detection by immunofluorescence as mentioned in the subsection “PLK-localization on chromosomes during pachytene is dependent on the SYP proteins”. Since PLK-2 is still present in the *spo-11* mutant and we see COSA-1::GFP foci on the pSYP-4 tracks, we reason that the single pSYP-4 tracks seen in *spo-11* mutants are due to CO precursor formation and PLK-2-dependent phosphorylation of SYP-4.

*5) Address concerns about chromosome movements within cells for FRAP experiments (subsection “FRAP Analysis”) in regards to FRAP analysis in Figure 6. Although the impact of cellular movement was minimized for intensity measurements, what about chromosomal movements within the cell? Meiotic prophase generally has been shown to display significant changes in motion of its chromosomes. It would be important to have data and a statement that this was not a concern in generating the change of fluorescent intensity. This is particularly important because z-projections were used for the analysis. Chromosome motion instead of incorporation of central element may or may not be distinguished well in projections.*

We now include a revised and expanded description in the Materials and methods section that also addresses the issue of chromosome movements. There we now explain that a small region of the nuclei was photobleached throughout the Z stack for the entire length of the nuclei and measured for extent of fluorescence recovery. We used the StackReg plugin in Fiji to correct for gross lateral animal movement and drift. Then we used rigid body registration in StackReg to minimize the impact of chromosomal movement on intensity measurements. In this case, because only the chromosomes are fluorescent (and not the nuclei as a whole) it is precisely the chromosomes themselves that are being registered over time. When this registration failed – for instance the chromosomal structure demonstrated obvious rotation around the lateral plane, and not just lateral translation, or rotation around the detection axis – then nuclei were excluded from further analysis due to our inability to reliably measure intensity of the same chromosomal region over time. Nuclei with the photobleached region (region of interest, ROI) in focus for the entire period of fluorescence recovery were selected for further analyses. To further correct for chromosomal movement, the photobleached regions were traced manually to measure average fluorescence intensity over the duration of the time-lapse.

We also include now two videos, one depicting an example of a wild type nucleus we scored and another showing a wild type nucleus we did not score due to poor chromosome registration over the duration of the time-lapse –Video 1 and Video 2.

*Major points that can be addressed by text fixes or changes to figures with available data:*

*6) Incorporate SC dynamics (and how they change) into the Figure 7 model. It is also suggested the authors should show the proposed feed-back loop, i.e. showing that DSBs trigger an event that in turn triggers their own switch off.*

As requested, we have made changes to the model on Figure 7 to incorporate SC dynamics and how they change. However, we respectfully choose to keep the current diagram where the negative inhibition on further DSB formation is shown on the chromosome at pachytene after phosphorylation of SYP-4 since our data supports a model in which PLK-1/2-dependent phosphorylation of SYP- 4 in response to CO designation results in changes in SC dynamics and that in turn switches off additional DSB formation along those chromosomes in pachytene.

*7) The authors are careful to point out in the Results section that their data are consistent with phosphorylation of SYP-3 being PLK dependent. However, they start their Discussion with: "We have discovered that phosphorylation of a central region component of the SC, SYP-4, by Polo-like kinases…" and "Our data indicate that Polo-like kinases phosphorylate SYP-4 starting at early pachytene…" The authors haven't provided any data demonstrating that PLK-1 or PLK-2 directly phosphorylates SYP-4. They can only say the phosphorylation is dependent on PLK-1/2 and that their data suggests that polo-like kinases phosphorylate SYP-4 in early pachytene, consistent with PLK-2's localization.*

We have now carefully gone over the text and made sure that we consistently only indicate that phosphorylation of SYP-4 is PLK-dependent. Examples of these changes are in the first paragraph of the Discussion.

*8) In Smolikov et al., PLOS Genetics (2009), elevated levels of persistent RAD-51 foci were interpreted as a defect in recombination progression, rather than as a failure to shut-off DSBs. Given the SYP-4 involvement in DSB shut-off, this might be a good opportunity to reevaluate the earlier findings. More generally, this would also be a good opportunity to reevaluate the implications of using rad-54 or other recombination-defective mutations as a tool to estimate total DSB levels in wild type.*

In Smolikov et al. (2009) we examined the levels of RAD-51 foci throughout the germline of *syp-4* null mutants. In that case, as in the Colaiacovo et al. (2003) and Smolikov et al. (2007) papers, the different *syp* null mutants (*syp-1, syp-2, syp-3* and *syp-4*) impaired the ability to stabilize homologous chromosome pairing interactions, but not initial DSB formation. As a result, DSBs formed but repair between homologs was not possible due to complete absence of an SC, resulting in persistent DSBs and a lack of crossovers. That is different from this current study in which in the context of the phosphodead and phosphomimetic mutants the SC proteins, such as SYP-4, are still associating with the chromosomes (Figure 6—figure supplement 1) and DSB repair can still take place between homologs resulting in detectable CO precursors (Figure 5). The use of a *rad-54* mutant to essentially “trap” RAD-51 at sites of DSB repair, is currently the only strategy available to assess DSB levels in the *C. elegans* germline. However, available data does argue that elevated RAD-51 can be due to more breaks, delayed repair, and/or increased time where breaks are occurring and therefore caution should be exercised when interpreting alteration of RAD-51 levels.

*9) Provide better introduction and discussion of current results relative to other organisms. Please introduce and discuss similarities and differences of DSB limitation via Plk in C. elegans compared to the role of ATM/Tel1 in other organisms. In addition, the discussion of similarities to the homolog engagement pathway for DSB regulation that has been previously documented in yeast and mouse needs to be improved. In the Introduction it is stated: "However, once a DSB is designated to become a CO a feedback mechanism turns off further programmed DSBs from forming (Lam and Keeney 2015). The molecular basis for transmission of this feedback regulation remains an open question." The first statement is not a fair summary of what is known. And the implication from second sentence seems to be that the current results will address the lack of information, but the connection of the current findings to these other organisms is not convincing. CO designation is one possibility for why "homolog engagement" turns off DSB formation in yeast and mouse, but it is not the only possibility. Keeney's and Toth's labs propose that it is SC formation that triggers DSB inhibition, in part because even non-homologous SC formation in the absence of SPO11 is accompanied by TRIP13-dependent HORMAD depletion from chromosomes in mouse (see Wojtasz 2009, Thacker 2014, and Keeney 2014). In any case, the text shouldn't state it so categorically that DSB inhibition is a consequence of CO designation, because there is certainly no evidence to strongly support that conclusion. Concerning the molecular basis of the feedback: indeed this is not known. But it clearly does not require polo kinase in yeast (Cdc5) because Keeney's lab showed that the homolog engagement (ZMM-dependent) pathway operates even in an ndt80 mutant (Thacker 2014). Cdc5 is not expressed in the absence of Ndt80. Minor point: the Lam and Keeney 2015 paper does not seem to be a particularly apt reference for this general point. Perhaps Keeney et al. 2014 would be better.*

As suggested, we modified our statement which now reads as follows: “However, the engagement of homologous chromosomes during recombination and/or SC assembly has been proposed to turn off further programmed DSB formation (reviewed in Keeney et al. 2014). One possibility is that once a DSB is designated to become a CO a feedback mechanism turns off further programmed DSBs from forming. An alternative, albeit non-mutually exclusive possibility, is that SC formation may result in structural changes along the chromosomes which suppress further DSB formation.” While we kept the second statement, that the molecular basis of the feedback is not known, we did modify the beginning of the following paragraph to clearly indicate that the findings regarding a polo kinase requirement apply to *C. elegans*. We also do this now in the first paragraph of the Discussion.

We also expanded our Discussion to address the similarities and differences of DSB limitation via PLK in *C. elegans* compared to the role of ATM/Tel1 in other organisms and expand on the point that it is SC formation that triggers DSB inhibition in yeast and mouse. Finally, we also replaced the Lam and Keeney (2015) for the Keeney et al. (2014) reference as recommended.

*10) Subsection “PLK-2 localization on chromosomes during pachytene is dependent on the SYP proteins”. In Figure 1, the localization in PLK-2 to the nuclear periphery of syp-4 is qualitatively different than in WT. In fact, PLK-2 location is different between syp-1 and syp-4 mutants. This is in contrast to the statement in the figure legend that PLK-2 localization is not different. So, either the figure for the syp-4 mutant is not representative or there is a difference that exists. There is also a distinct difference in the HTP-3 staining for syp-4. Some comment should be made.*

In this figure we are showing that PLK-2 still forms aggregates, likely at the pairing centers, during the leptotene/zygotene stage in *syp-1* and *syp-4* mutants, but that we no longer see any PLK-2 signal on pachytene stage nuclei. However, in the *syp-1* and *syp-4* mutants, as previously reported in MacQueen et al. (2002) and Smolikov et al. (2009), while chromosomes initially attempt to pair at leptotene/zygotene, the levels of pairing at that stage are somewhat lower than in wild type. Therefore, the pairing centers of chromosomes are not held in close juxtaposition at the same levels as in wild type. This is why the PLK-2 signal forms fewer and larger aggregates in wild type (where homologs find each other more effectively and start to stabilize those pairing interactions) compared to the *syp* null mutants at that stage (where homologs pair less frequently and do not successfully stabilize those interactions). We now explain this in the figure legend for Figure 1. Moreover, we replaced the *syp-1* and *syp-4* mutant images in Figure 1 for more representative images that show the higher number of smaller PLK-2 aggregates observed at leptotene/zygotene in both mutants compared to wild type. The nuclei shown for wild type, *syp-1*, and *syp-4* mutants are all from the same stage.

HTP-3 staining is different in wild type compared to both *syp-1* and *syp-4* null mutants because chromosomes are synapsed in wild type and therefore their axes are held in close juxtaposition, which is observed as thicker HTP-3 signal, while in the *syp* null mutants homologous chromosomes are not held together and a lot more thinner tracks of HTP-3 are observed. We also explain this now in the figure legend.

*11) In Figure 1, the image contrast/ detection thresholds for WT and syp mutants appear to be set differently. If anything, the background should be set lighter in syp mutants. The same concern applies for PLK-1 in Figure 1—figure supplement 1, and Figure 2 the syp-4 and plk-2 panels. In Figure 3, the colored boxes are hard to see and the fact that different antibodies and stages are identically color-coded is confusing (e.g. green stands for COSA-1 and also midpachytene).*

We have now corrected this and have set the same image intensity for the images in Figure 1, Figure 1—figure supplement 1 and Figure 2. In Figure 3, we removed the colored boxes and included high-resolution images of single nuclei from early, mid and late pachytene stages to more clearly show the colocalization of COSA-1::GFP foci with pSYP-4.

*12) Primary data should be provided for RAD-51 foci in the S269A mutant (i.e. a representative IF image).*

We now show a representative IF image in Figure 5—figure supplement 1.

*13) The possible cause/effect relationship between SC dynamics and extra RAD-51 should be discussed clearly. While the inference is that continued SC dynamics result in extra DSBs, currently there is no evidence to exclude the reverse relationship. If additional experimental data are available, great; if not, adding discussion would suffice. For example, have the authors explored whether the SC in syp-4 (A269S) still exhibits high dynamics in the absence of DSBs/SPO-11? If syp-4 (S269A)-induced SC dynamics persist despite absence of DSBs, this would allow excluding a role of extra DSBs in inducing SC dynamic. Or: does the S269D phosphomimetic mutation eliminate SC dynamics in absence of DSBs? Notably, even in absence of DSBs, some SYP-4 phosphorylation is still happening. Is it possible to compare the SC dynamics phenotypes of S269A and other recombination mutant(s) that assemble apparently normal SC?*

We propose that the SC is stabilized by PLK-dependent phosphorylation, which inhibits further DSB formation. In the absence of PLK-dependent SYP-4 phosphorylation in *syp-4(S269A)* phosphodead mutants, the SC persists in a more dynamic state in mid-pachytene and chromosomes exhibit significantly elevated numbers of DSBs. Consistent with our model, in *spo-11* mutants, which we showed only have <20% of nuclei with pSYP-4 signal along a single chromosome, the SC persisted in a more dynamic state throughout pachytene as measured by FRAP analysis in Machovina et al., 2016 and Pattabiraman and Villeneuve (personal communication). This rules out the argument that it is the extra DSBs that cause the SC to remain in a more dynamic state. We now better address this in the first paragraph of the Discussion.

*14) Does SYP-4 phos work via chromosome association of DSB-1 and -2? It would be interesting to discuss this possibility.*

To address this comment we examined pSYP-4 localization in *dsb-1 and dsb-2* mutants. Similar to what we observed for other mutants defective in early steps of recombination we did not detect pSYP-4 signal in most of the germline nuclei in *dsb-1 and dsb-2* mutants and only a few nuclei showed a pSYP-4 track (Figure 3). This new result is described in the subsection “Phosphorylation of SYP-4 is dependent on crossover precursor formation” and discussed in the first paragraph of the Discussion.

*15) It seems that the authors' model SYP-4 phos should be more important when total DSBs are low (e.g., in a dsb-2 mutant or spo-11 + low IR dose). Did the authors test this? If not, it could be a Discussion point.*

Our model is that SYP-4 is phosphorylated on the chromosomes that undergo interhomolog CO precursor formation. In the absence of CO formation, for example in pro-crossover mutants, although they undergo DSB formation we do not detect SYP-4 phosphorylation. In addition, in *spo-11(ok79)* mutant animals expressing COSA-1::GFP, the pSYP-4 tracks colocalize with COSA-1::GFP foci (Figure 3—figure supplement 3). This argues that even if a chromosome might undergo only one DSB, as long as this break gets processed into a CO you will get phosphorylated SYP-4 along that chromosome.

As described for point #14, we examined pSYP-4 localization in *dsb-1* and *dsb-2* mutants, and similar to other recombination defective mutants, we did not detect pSYP-4 signal in most nuclei and only a few nuclei carried a pSYP-4 track (we added this new data to Figure 3). Extrapolating from our *spo-11* mutant analysis, we hypothesize that those chromosomes with the pSYP-4 signal in *dsb- 1* and *dsb-2* mutants also correspond to chromosome that underwent a DSB and CO precursor formation.